# Optogenetic stimulation of the liver-projecting melanocortinergic pathway promotes hepatic glucose production

Eunjin Kwon [1,2], Hye-Young Joung [1,2], Shun-Mei Liu[1,2], Streamson C. Chua Jr.[1,2], Gary J. Schwartz [1,2] & Young-Hwan Jo [1,2,3 ✉]

The central melanocortin system plays a fundamental role in the control of feeding and body weight. Proopiomelanocortin (POMC) neurons in the arcuate nucleus of the hypothalamus (ARC) also regulate overall glucose homeostasis via insulin-dependent and -independent pathways. Here, we report that a subset of ARC POMC neurons innervate the liver via preganglionic parasympathetic acetylcholine (ACh) neurons in the dorsal motor nucleus of the vagus (DMV). Optogenetic stimulation of this liver-projecting melanocortinergic pathway elevates blood glucose levels that is associated with increased expression of hepatic gluco-neogenic enzymes in female and male mice. Pharmacological blockade and knockdown of the melanocortin-4 receptor gene in the DMV abolish this stimulation-induced effect. Activation of melanocortin-4 receptors inhibits DMV cholinergic neurons and optogenetic inhibition of liver-projecting parasympathetic cholinergic fibers increases blood glucose levels. This elevated blood glucose is not due to altered pancreatic hormone release. Interestingly, insulin-induced hypoglycemia increases ARC POMC neuron activity. Hence, this liver-projecting melanocortinergic circuit that we identified may play a critical role in the counterregulatory response to hypoglycemia.

[1] The Fleischer Institute for Diabetes and Metabolism, Albert Einstein College of Medicine, Bronx, NY, USA. [2] Division of Endocrinology, Department of Medicine, Albert Einstein College of Medicine, Bronx, NY, USA. [3] Department of Molecular Pharmacology, Albert Einstein College of Medicine, Bronx, NY, USA. ✉email: young-hwan.jo@einsteinmed.org

The liver plays an important role in maintaining glucose homeostasis. Hepatic glucose production results from de novo synthesis via gluconeogenesis and/or from degradation of hepatic glycogen via glycogenolysis. The central melanocortin system regulates this process in insulin-dependent and/or -independent means[1–10]. For instance, defective α-melanocyte-stimulating hormone (α-MSH) production promotes endogenous gluconeogenesis[1,10]. Chemogenetic activation of the whole population of POMC neurons in the arcuate nucleus of the hypothalamus (ARC) decreases hepatic glucose production[3], whereas reduced POMC neuron activity in mice lacking p70 ribosomal S6 protein kinase-1 causes increased hepatic glucose production without altering food intake[4]. Activation of ARC POMC neurons by glucagon-like peptide 2 and leptin increases hepatic insulin sensitivity[2,9]. Additionally, transgenic neuronal expression of POMC in leptin-deficient obese mice normalizes hyperglycemia and glucose intolerance by regulating glucose metabolism in liver[11]. Hypothalamic POMC rescue in obese mice also attenuates hyperglycemia, hyperinsulinemia, and hepatic steatosis[7]. This regulation is likely to be mediated by neural connections between ARC POMC neurons and the autonomic nervous system, in particular the sympathetic nervous system[5,12,13].

However, there still exist foundational gaps in our knowledge of the neurobiology and neuroanatomy of the central melanocortin system that regulates hepatic glucose production. In fact, intracerebroventricular (ICV) administration of the melanocortin-4 receptor (MC4R) agonist melanotan II (MTII) increases rather than decreases blood glucose levels in rodents[14,15], which is in contrast with the findings with POMC transgenic and mutant mice described above[3–9,11,16]. More surprisingly, mice lacking the Pomc gene exclusively in the hypothalamus show improved glucose tolerance and normal fasting glycemia, while mutant animals develop severe obesity and insulin resistance[5]. Recently, it has been also documented that long-term inhibition of ARC POMC neurons reduces blood glucose levels[17]. These prior findings suggest that the central melanocortin system may have the ability to promote endogenous glucose production. Indeed, a recent study with mice lacking the Pomc gene exclusively in the hypothalamus shows impaired counterregulatory responses to hypoglycemia[13]. This hypoglycemia counterregulation is mediated in part through activation of POMC neurons in the ARC and MC4R-expressing neurons in the paraventricular nucleus of the hypothalamus (PVN)[13], further suggesting the involvement of the sympathetic nervous system.

It has been well documented that POMC neurons in the ARC innervate liver[18] and that ARC POMC neurons send axonal projections to both the dorsal motor nucleus of the vagus (DMV) and the intermediolateral cell column of the spinal cord (IML) via direct connection[19,20]. Interestingly, treatment with MC4R agonists excites preganglionic sympathetic cholinergic neurons in the IML, but unexpectedly inhibits preganglionic parasympathetic acetylcholine (ACh)-expressing neurons in the DMV[21]. In other words, melanocortins are able to oppositely regulate parasympathetic and sympathetic outflow. In this study, we specifically examine the role of the $ARC^{POMC} \rightarrow DMV^{ACh} \rightarrow$ liver projection in the regulation of hepatic glucose production. We demonstrate that acute optogenetic stimulation of the $ARC^{POMC} \rightarrow DMV^{ACh} \rightarrow$ liver projection upregulates mRNA expression for gluconeogenic enzymes in the liver and elevates blood glucose levels through inhibition of parasympathetic cholinergic outflow to liver. In addition, insulin-induced hypoglycemia increases ARC POMC neuron activity, supporting the interpretation that ARC POMC neurons may play a critical role in the counterregulatory response to hypoglycemia.

## Results

### A subset of POMC neurons in the ARC projects to liver via DMV cholinergic neurons. We first examined whether ARC

POMC neurons send projections to liver via DMV cholinergic neurons, as the DMV contains preganglionic parasympathetic cholinergic neurons innervating liver[22]. We injected a Cre-dependent anterograde AAV1-CAG-FLEX-GFPsm-myc (AAV1-FLEX-GFPsm) virus[23] into the ARC of POMC-Cre mice. The ARC of POMC-Cre mice injected with AAV1-FLEX-GFPsm viruses exhibited GFP-positive cells, representing POMC neurons (Fig. 1a). In the same animals, GFP-positive fibers were also detected in the DMV (Fig. 1b and Supplementary Fig. 1b). Immunostaining with an anti-choline acetyltransferase (ChAT), a cholinergic neuronal marker, antibody further revealed that DMV cholinergic neurons received POMC input from the ARC (Fig. 1b and Supplementary Fig. 1b), consistent with the prior finding that the DMV is a downstream target of ARC POMC neurons[19]. Immunolabeling with an anti-MC4R antibody further revealed dense expression of MC4Rs in the DMV (Supplementary Fig. 1c). In our neurophysiological analysis of DMV cholinergic neurons, treatment with the MC4R agonist MTII (100 nM) significantly reduced the mean frequency of action potentials in 70% of the neurons examined (Fig. 1c, d), consistent with the prior study of Elmquist and colleagues[21] describing that activation of MC4Rs suppresses the activity of DMV cholinergic neurons.

In order to identify liver-projecting POMC neurons in the ARC and to control them with optogenetics, we used an AAV serotype 8 (AAV8) encoding a fusion protein of wheat-germ agglutinin (WGA) and Cre recombinase (AAV8-EF1α-IRES-WGA-Cre (AAV8-WGA-Cre))[24] (Fig. 2). This AAV produces transneuronal trafficking of the WGA–Cre fusion protein[24]. To determine transneuronal transport of this fusion protein and its Cre recombinase activity, we directly injected AAV8-WGA-Cre into the liver of floxed-stop Rosa26-GFP mice (Fig. 2a). At 3−4 weeks post viral injections, we conducted immunolabeling with an anti-GFP antibody and found that the DMV exhibited Cre-mediated GFP expression (Fig. 2b). We then asked whether these GFP-positive neurons are preganglionic parasympathetic cholinergic neurons in the DMV. Immunostaining with an anti-ChAT antibody revealed that about half of ChAT-positive cells were positive for GFP (Fig. 2b, 48.2 ± 2.6%, $n = 5$ mice) and almost all the GFP-expressing neurons were stained with an anti-ChAT antibody (98 ± 0.5%, $n = 5$ mice), indicating that GFP-positive cells are liver-projecting preganglionic parasympathetic cholinergic neurons. We next examined whether the Cre transgene reaches to the ARC (Fig. 2c). Immunolabeling with an anti-GFP antibody showed Cre-mediated GFP expression in the ARC (Fig. 2d), further supporting the validity of our experimental approach. We found that approximately 20% of POMC-positive cells were co-stained with an anti-GFP antibody (Fig. 2d; $n = 5$ mice), consistent with the prior work describing that a small subset of ARC POMC neurons innervate liver[18].

Among retrogradely identified liver-innervating neurons in the brain, we investigated the role of ARC neurons that project to the liver. To exclusively control the activity of these liver-projecting POMC neurons, we administered an AAV5 encoding Cre-dependent channelrhodopsin-2 (ChR2) and enhanced yellow fluorescent protein (eYFP) fusion protein under control of the two neuronal POMC enhancers (nPEs)[25] (AAV5-nPE-DIO-ChR2-YFP) into the ARC of C57BL/6J wild-type (WT) mice as documented in our recent work[26]. At 4–6 weeks prior to these viral injections into the ARC, AAV8-WGA-Cre viruses were directly injected into the livers of WT mice (Fig. 2e). Under these experimental conditions, we found that the ARC of these mice had YFP-positive cells (Fig. 2f and Supplementary Fig. 2), representing Cre-mediated ChR2-expressing cells. Double immunostaining with anti-POMC and anti-YFP antibodies showed co-expression of POMC (Fig. 2f and Supplementary Fig. 2). We further examined whether these liver-projecting POMC neurons send direct

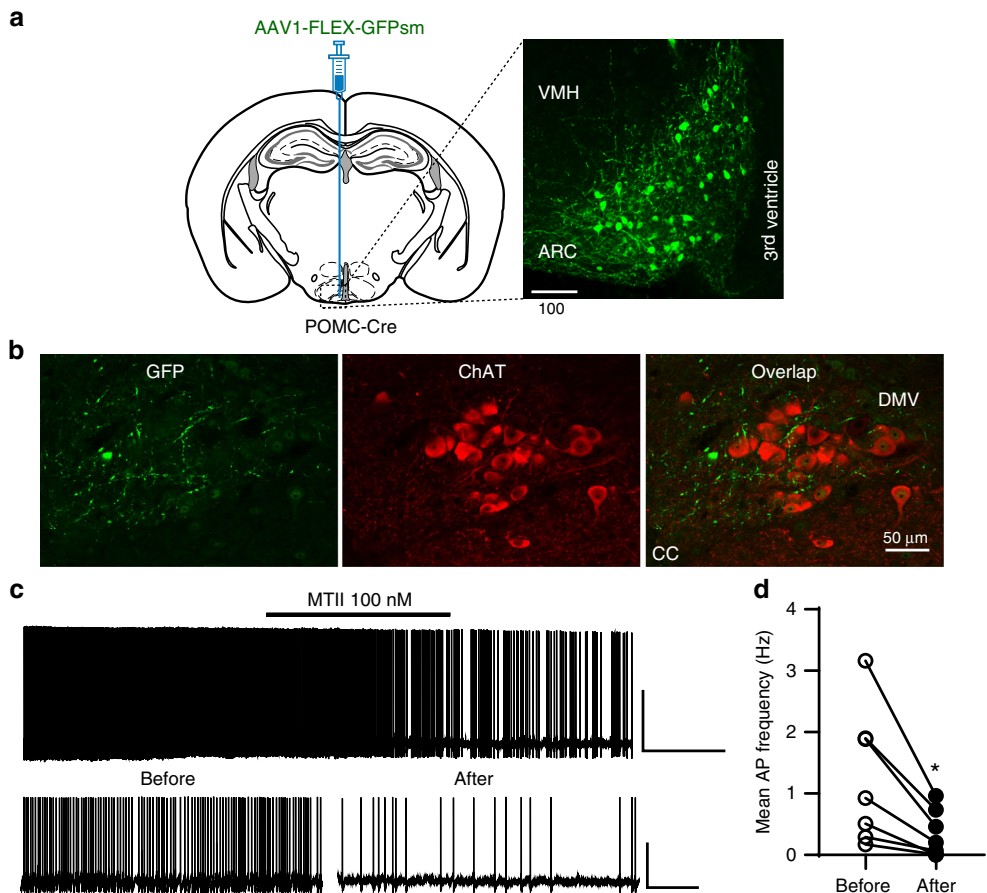

**Fig. 1 DMV cholinergic neurons receive POMC input from the ARC. a** Schematic diagram of the experimental configuration. AAV1-FLEX-GFPsm viruses were injected into the ARC of POMC-Cre mice (left panel). Expression of GFP-positive cells was clearly visible in in the ARC (right panel). Scale bar: 100 µm. VMH ventromedial hypothalamus. **b** Images of confocal fluorescence microscopy showing double immunostaining with anti-GFP (green, left panel) and anti-ChAT (red, middle panel) antibodies. GFP-positive fibers were visible in the DMV (right panel), representing DMV cholinergic neurons received POMC input from the ARC. Scale bar: 50 µm. CC central canal. **c, d** Representative traces showing reduced DMV cholinergic neuron activity in response to treatment with MTII (100 nM) in the absence of synaptic blockers (**c**). Plots showing individual data as to mean frequency of action potentials before and after treatment with MTII (**d**) (mean frequency: 1.3 ± 0.3 Hz vs. 0.3 ± 0.1 Hz, $n = 7$ out of ten neurons, two-tailed $t$ test, *$p = 0.02$; $V_{\mathrm{m}}$: −45.6 ± 1.8 mV vs. −45.8 ± 2.2 mV, two-tailed $t$ test, $p = 0.9$). All data are shown as mean ± SEM. Source data are provided as a Source Data file.

projections to DMV cholinergic neurons as shown in Fig. 1b. Double immunostaining with anti-YFP and ChAT antibodies revealed that DMV ChAT neurons were surrounded with YFP-positive axons originating in the ARC (Fig. 2g, h), supporting the interpretation that DMV cholinergic neurons received synaptic input from liver-projecting POMC neurons in the ARC. Thus, we are able to efficiently deliver the Cre transgene from the liver to the ARC and to express light-activated ChR2 exclusively in liver-projecting ARC POMC neurons.

**Optogenetic stimulation of the ARC$^{POMC}$ → DMV$^{ACh}$ → liver projection elevates blood glucose levels.** We next examined whether stimulation of POMC input to the DMV alters blood glucose levels. We optogenetically stimulated POMC fibers in the DMV of the WT mice injected with AAV8-WGA-Cre in the liver and AAV5-nPE-DIO-ChR2-YFP in the ARC (Fig. 3a). Optogenetic stimulation of this ARC$^{POMC}$ → DMV$^{ACh}$ → liver projection at 20 Hz for 1 h significantly elevated blood glucose levels in both female and male mice (Fig. 3b–e). In contrast, mice with off-target implantation of a fiber-optic cannula and missed viral injections exhibited no changes in blood glucose levels by the end of the 1-h optogenetic stimulation (Supplementary Fig. 3a–e). Additionally, as high light powers cause heating and reduce the

firing rate[27], we examined a possible thermal effect in mice injected with control viruses only and found that light illumination did not cause an increase in blood glucose levels (Supplementary Fig. 3f–h). The onset of the glucose-elevating effects of optogenetic stimulation was approximately 15 min and this effect rapidly returned to baseline after optogenetic stimulation (Fig. 3b, d). Blood glucose levels increased from 98.4 ± 6.5 to 155.1 ± 13.9 mg/dl in males and from 70.8 ± 5.9 to 113.6 ± 8.7 mg/dl in females by the end of the 1-h optogenetic stimulation period (Fig. 3c, e). To determine the involvement of MC4Rs in these responses, we intraperitoneally injected the MC4R antagonist SHU9119 (100 µg/kg). As shown in Fig. 3f, treatment with SHU9119 completely abolished the ability of optogenetic stimulation to elevate blood glucose levels.

We then asked whether this increase in blood glucose levels is due in part to improved hepatic glucose production. Liver tissues were harvested from mice with and without the 1-h optogenetic stimulation at 20 Hz and qPCR analysis was carried out to measure the expression of key hepatic gluconeogenic genes such as glucose-6-phosphatase (G6Pase) that is an enzyme hydrolyzing glucose-6-phosphate, resulting in the creation of glucose and phosphoenolpyruvate carboxykinase (PEPCK) that converts oxaloacetate into phosphoenolpyruvate that is a key step in gluconeogenesis (Supplementary Table 1). Both *G6pase* and

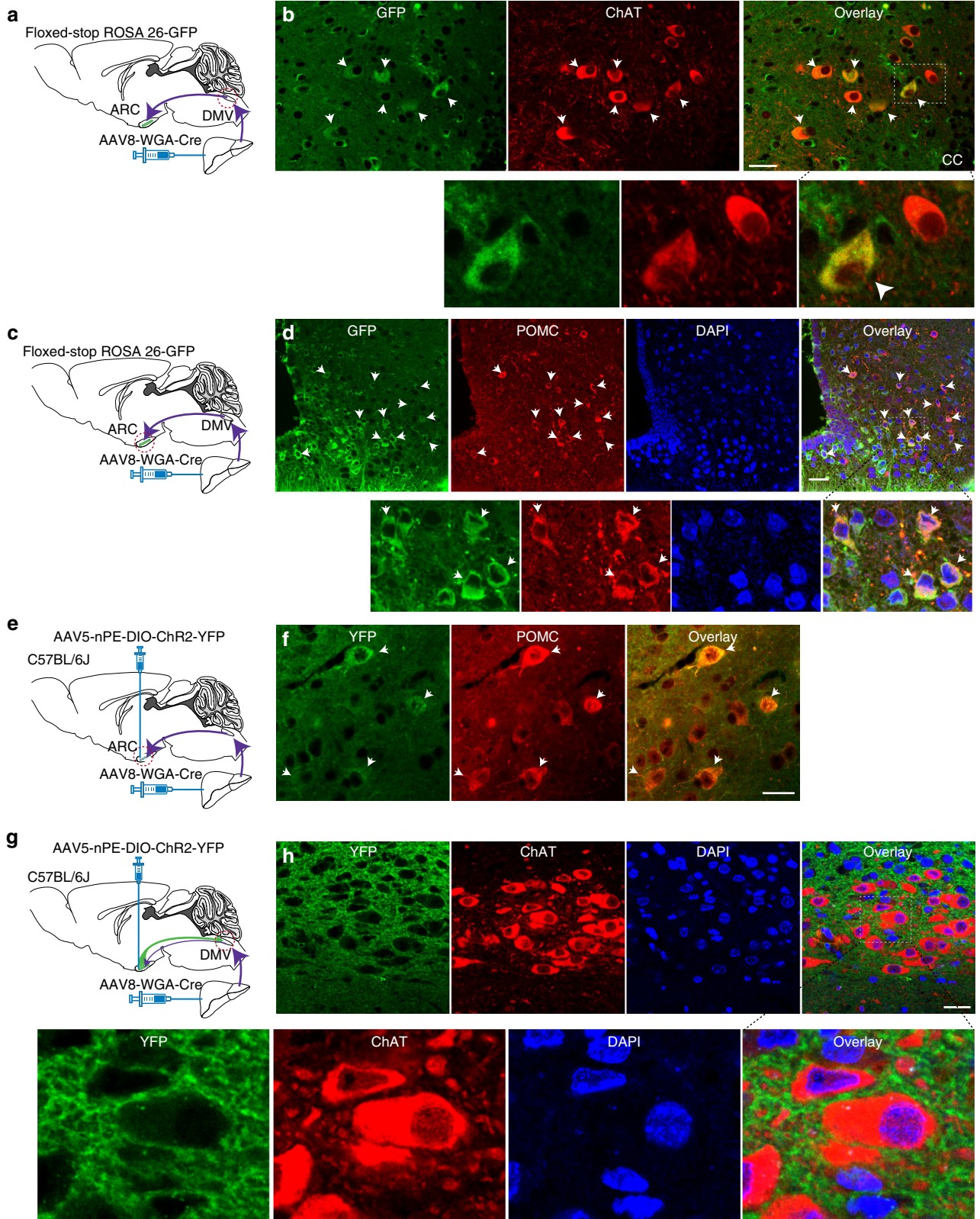

*Pepck* gene expression was significantly higher in mice with optogenetic stimulation than in mice without light illumination (Fig. 3g). As these changes in the gene expression can be due to the release of pancreatic hormones, we investigated whether there is altered pancreatic hormone secretion. Immediately after the 1-h optogenetic stimulation, blood samples were collected from the retroorbital plexus and plasma glucagon and insulin concentrations were measured. We found that there was no significant change in plasma glucagon levels (Fig. 3h), supporting the interpretation that this increased blood glucose levels is not due to glucagon secretion. In line with this finding, plasma insulin levels were not significantly different between the two groups, although there was a trend toward an increase (Fig. 3i). Furthermore, this stimulation did not significantly alter plasma

**Fig. 2 ARC POMC neurons innervate the liver via DMV cholinergic neurons. a, b** Schematic diagram of the experimental configuration. AAV8-WGA-Cre viruses were injected into the liver of floxed-stop Rosa26-GFP mice (**a**). (**b**) Images of confocal fluorescence microscopy showing co-expression of GFP (left panel) and ChAT (middle panel) in a subset of DMV cholinergic neurons (right panel, white arrows). GFP-positive cells in the DMV represent retrogradely identified neurons. Bottom panel: higher magnification view of neurons co-expressing GFP (left) and ChAT (middle) (right panel, white arrowhead). Scale bar: 30 μm. CC central canal. **c, d** Schematic illustration of our experimental configuration (**c**). (**d**) Images of confocal fluorescence microscopy showing Cre-mediated GFP expression in the ARC. A subset of GFP-positive cells (left panel) co-expressed POMC (red, middle panel) (right panel, white arrows). Bottom panel: higher magnification view of neurons co-expressing GFP and POMC (right panel, white arrow; green, GFP; red, POMC; blue, DAPI). Scale bar: 30 μm. **e, f** Schematic diagram of our experimental configuration. AAV8-WGA-Cre viruses were injected into the liver prior to injection of AAV5-nPE-DIO-ChR2-eYFP viruses into the ARC of C57BL/6J mice (**e**). (**f**) Images of confocal fluorescence microscopy showing that a subset of POMC (middle panel) neurons expressed ChR2 (right panel, white arrows). Green, YFP staining. Scale bar: 20 μm. **g, h** Images of confocal fluorescence microscopy showing double immunostaining with anti-YFP (green) and anti-ChAT (red) antibodies. DMV ChAT neurons received YFP-positive axon terminals from the ARC (right panel), indicating that ARC POMC neurons innervate liver through DMV cholinergic neurons. Bottom panel: higher magnification view of DMV cholinergic neurons (red) receiving YFP-positive fibers (green) and terminals. Scale bar: 25 μm.

concentrations of the sympathetic adrenal hormone corticosterone (Fig. 3j). Hence, our results are consistent with the interpretation that stimulation of the $ARC^{POMC} \rightarrow DMV^{ACh} \rightarrow$ liver projection elevates blood glucose levels at least in part via increased gluconeogenic enzyme expression.

We further performed a pyruvate tolerance test (PTT) as hepatic gluconeogenesis can be estimated using PTT that measures systemic elevation of glucose derived from pyruvate. WT mice injected with AAV8-WGA-Cre in the liver and AAV5-nPE-DIO-ChR2-YFP into the ARC were fasted overnight (15 h) prior to the start of experiment (Fig. 4a). Pyruvate (2 g/kg body weight) was intraperitoneally injected and circulating glucose concentrations were measured at 0, 15, 30, 45, 60, 90, and 120 min (Fig. 4b, c). Optogenetic stimulation of the $ARC^{POMC} \rightarrow DMV^{ACh} \rightarrow$ liver projection significantly elevated blood glucose levels compared to control mice without optogenetic stimulation (Fig. 4b, c), further supporting that activation of this pathway promotes hepatic glucose output. To examine a contribution of MC4Rs in these responses, SHU9119 was i.p. injected 30 min prior to optogenetic stimulation. Treatment with SHU9119 abolished the effect of the glucose-elevating effects of optogenetic stimulation (Fig. 4d, e). To further test the involvement of MC4Rs expressed in the DMV, we knock down the *Mc4r* gene by injecting *Mc4r* shRNAs into the dorsal vagal complex (DVC) (Supplementary Fig. 4a and Supplementary Table 1). Under these experimental conditions, stimulation of the $ARC^{POMC} \rightarrow DMV^{ACh} \rightarrow$ liver projection failed to increase blood glucose levels in mice injected with pyruvate (Fig. 4e). In addition, glucose tolerance tests showed that WT mice with stimulation also exhibited higher glucose levels than WT mice without optogenetic stimulation (Supplementary Fig. 5a, b). Thus, our results are consistent with the early study of Rossetti and colleagues describing the stimulatory effect of central MC4Rs on hepatic glucose production via gluconeogenesis[15].

**Inhibition of parasympathetic cholinergic neurons elevates blood glucose levels.** Despite the importance of the parasympathetic regulation of glucose homeostasis[28,29], the involvement of preganglionic parasympathetic cholinergic neurons in hepatic glucose metabolism remains uncertain, as mice lacking the muscarinic acetylcholine receptor (mAChR) type 3 in hepatocytes exhibit no metabolic differences[30]. To examine the role of parasympathetic cholinergic outflow on hepatic glucose output in mice, we cut the hepatic branch of the vagus. Hepatic branch vagotomy caused hyperglycemia under basal conditions (Supplementary Fig. 6a), consistent with the prior finding with hepatic-vagotomized rodents[31]. Importantly, optogenetic stimulation of the $ARC^{POMC} \rightarrow DMV^{ACh} \rightarrow$ liver projection was no longer able to elevate blood glucose levels in hepatic-vagotomized mice (Supplementary Fig. 6b, c).

As activation of MC4Rs in DMV cholinergic neurons inhibits their activity (Fig. 1c) and stimulation of POMC innervation elevates blood glucose levels (Fig. 3), we investigated whether reduced parasympathetic cholinergic outflow is able to regulate blood glucose levels as well. To control liver-projecting cholinergic nerve activity, we expressed inhibitory light-activated opsins exclusively in parasympathetic cholinergic fibers by injecting Cre-dependent retrograde viral vectors (retroAAV2-FLEX-Jaws-GFP) into the livers of ChAT-IRES-Cre mice (Fig. 5a). In the DMV of these ChAT-IRES-Cre mice, there were GFP-positive neurons that represent inhibitory Jaws-expressing cells (Fig. 5a). As shown in Fig. 2b, only a subset of cholinergic neurons expressed GFP (Fig. 5a). Importantly, immunolabeling with an anti-GFP antibody revealed that hepatocytes in these mice were innervated with GFP-positive fibers, indicating that Jaws-expressing cholinergic neurons in the DMV project to hepatocytes (Fig. 5a). These GFP-positive fibers were also stained with an anti-ChAT antibody (Supplementary Fig. 7a). Additionally, we carried out qPCR analysis of mAChR expression in liver (Supplementary Table 1). We found that liver cells expressed mAChRs (M1-M5), but mAChR type 5 appeared to be less abundant compared to other types of mAChRs (Supplementary Fig. 7b).

To examine the role of parasympathetic cholinergic innervation in hepatic gluconeogenesis, SMD-LEDs (wavelength, 620 nm) were used to activate inhibitory Jaws with 1 Hz light stimulation (1 s pulse duration, 1 pulse per 1 s, 3 s interval, 1 h duration, Fig. 5b). This noninvasive optogenetic stimulation method effectively alters autonomic nerve activity in our prior study[32]. To validate our experimental approach, we measured ACh content in the liver. We found a significant reduction in ACh content in the livers of mice injected with Jaws (Fig. 5c), further supporting that light illumination inhibits cholinergic fibers. Much to our surprise, silencing liver-projecting cholinergic fibers augmented blood glucose levels in both female and male mice (Fig. 5d–g). The onset of the effect was similar to that obtained with stimulation of the $ARC^{POMC} \rightarrow DMV^{ACh} \rightarrow$ liver projection. These results support the interpretation that reduced activity of liver-projecting parasympathetic cholinergic neurons alone is able to increase hepatic glucose output.

We carried out qPCR analysis to examine whether this inhibition upregulates hepatic *G6pase* and *Pepck* mRNA expression. Light illumination for 1 h significantly upregulated *G6pase* and *Pepck* mRNA expression in ChAT-IRES-Cre mice with Jaws compared with ChAT-IRES-Cre mice without Jaws (Fig. 5h, i). This increase was not due to increased glucagon release as there was no significant change in plasma glucagon levels (Fig. 5j). Interestingly, plasma insulin levels were significantly higher in ChAT-IRES-Cre mice expressing Jaws in cholinergic nerves than in ChAT-IRES-Cre mice (Fig. 5k). This may be attributable to glucose-stimulated

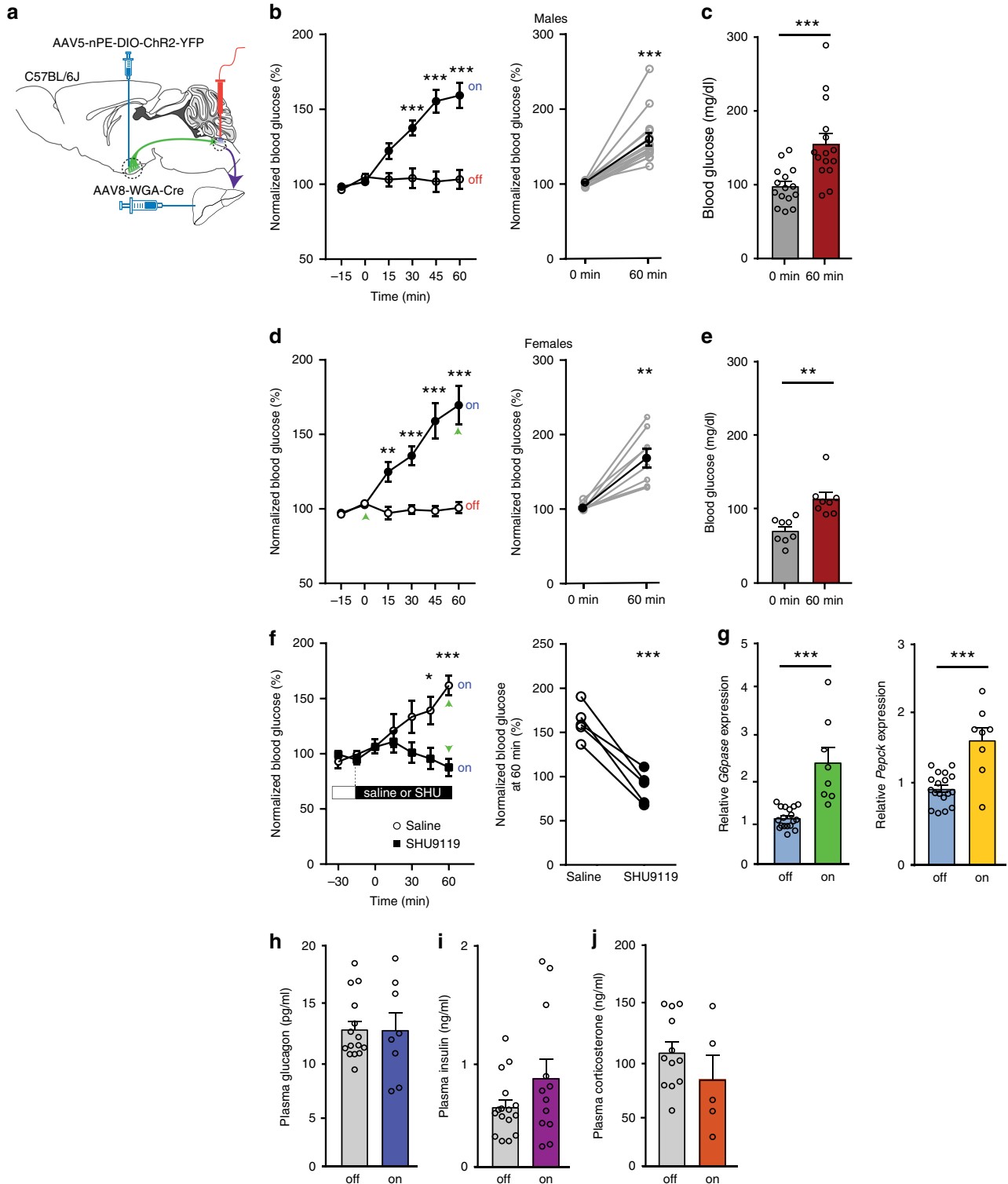

insulin secretion (GSIS) as inhibition of liver-projecting cholinergic nerves significantly elevated blood glucose that may activate the counterregulatory response. There were no significant differences in plasma epinephrine and corticosterone levels between the two groups (Supplementary Fig. 8a, b). Inhibition of parasympathetic cholinergic input did not change plasma lactate levels in both male and female mice (Supplementary Fig. 8a, b). Therefore, our results support the interpretation that reduced activity of liver-projecting cholinergic neurons elevates blood glucose levels in a pancreatic hormone-independent manner.

**Insulin-induced hypoglycemia excites POMC neuronal activity.** A recent study has described that ARC POMC neurons are required for the normal counterregulatory response to hypoglycemia[13]. Interestingly, contrary to in vitro neurophysiological responses to glucose[8], insulin-induced hypoglycemia and 2-deoxyglucose-mediated glucopenia each induce c-fos mRNA expression in ARC POMC neurons in vivo[13], suggesting that ARC POMC neuronal activity increases in response to hypoglycemia, possibly due to altered synaptic strength at ARC POMC neuronal synapses. We thus measured ARC POMC neuronal

**Fig. 3 Optogenetic stimulation of the ARC$^{POMC}$ → DMV$^{ACh}$ → liver projection causes hepatic gluconeogenesis. a** Schematic diagram of the experimental configuration. AAV8-WGA-Cre viruses were injected into the liver and 2−3 weeks post viral injections, AAV5-nPE-ChR2-YFP viruses were injected into the ARC of C57BL/6J mice. The ARC$^{POMC}$ → DMV$^{ACh}$ → liver projection was optogenetically stimulated at 20 Hz for 1 h. **b**, **c** Pooled data showing that optogenetic stimulation of the ARC$^{POMC}$ → DMV$^{ACh}$ → liver projection increased blood glucose levels in male mice (control (laser off), $n = 12$ mice (open circle), stimulation (laser on), $n = 15$ mice (filled circle)), two-way RM ANOVA followed by Sidak multiple comparisons test, interaction $F_{(5, 125)} = 18.2$, $p < 0.001$, time $F_{(5, 125)} = 21.4$, $p < 0.001$, between the groups $F_{(1, 25)} = 24.1$, $p < 0.001$; 30 min, ***$p < 0.001$; 45 min, ***$p < 0.001$; 60 min, ***$p < 0.001$ (**b**, left panel). Right panel: pooled data showing increased blood glucose levels at the end of optogenetic stimulation (two-tailed $t$ test, ***$p < 0.001$, vs. 0 min, $n = 15$ mice). **c** Pooled data showing changes in blood glucose levels before and after 1-h optogenetic stimulation (two-tailed $t$ test, ***$p < 0.001$, vs. 0 min, $n = 15$ mice). Mice were not fasted. All data are shown as mean ± SEM. Source data are provided as a Source Data file. **d**, **e** Pooled data showing effect of optogenetic stimulation of the ARC$^{POMC}$ → DMV$^{ACh}$ → liver projection on blood glucose in female mice (control (laser off), $n = 10$ mice (open circle), stimulation (laser on), $n = 8$ mice (filled circle)), two-way RM ANOVA followed by Sidak multiple comparisons test, interaction $F_{(5, 80)} = 18.5$, $p < 0.001$, time $F_{(5, 80)} = 18.9$, $p < 0.001$, between the groups $F_{(1, 16)} = 38.6$, $p < 0.001$; 15 min, **$p = 0.005$; 30 min, ***$p < 0.001$; 45 min, ***$p < 0.001$; 60 min, ***$p < 0.001$ (**d**, left panel). Right panel: summary plot showing increased blood glucose levels before and after optogenetic stimulation (two-tailed $t$ test, **$p = 0.001$, vs. 0 min, $n = 8$ mice). **e** Pooled data showing changes in blood glucose levels before and after 1-h optogenetic stimulation (two-tailed $t$ test, **$p = 0.001$ vs. 0 min, $n = 8$ mice). Mice were not fasted. All data are shown as mean ± SEM. Source data are provided as a Source Data file. **f** Pooled data showing effect of SHU9119 on optogenetic stimulation-induced increase in blood glucose (saline, $n = 5$ mice, SHU9119, $n = 5$ mice, two-way RM ANOVA followed by Sidak multiple comparisons test, interaction $F_{(5, 40)} = 8.6$, $p < 0.001$, time $F_{(5, 40)} = 4.2$, $p < 0.01$, between the two groups, $F_{(1, 8)} = 7.5$, $p = 0.03$; 45 min, *$p = 0.01$, 60 min, ***$p < 0.001$, left panel). Right panel: summary plot showing changes in blood glucose at the end of optogenetic stimulation with saline or SHU9119 (two-tailed $t$ test, ***$p < 0.001$, vs. saline, $n = 5$ mice). Mice were not fasted. All data are shown as mean ± SEM. Source data are provided as a Source Data file. **g** Pooled data showing increased *G6pase* and *Pepck* mRNA expression in mice with optogenetic stimulation (two-tailed $t$ test, ***$p < 0.001$, vs. control, control, $n = 18$ mice; stimulation, $n = 8$ mice). All data are shown as mean ± SEM. Source data are provided as a Source Data file. **h–j** Summary plots showing plasma glucagon (**h**), insulin (**i**), and corticosterone (**j**) levels following stimulation of the ARC$^{POMC}$ → DMV$^{ACh}$ → liver projection (two-tailed $t$ test, glucagon, control, $n = 15$ mice, stimulation, $n = 8$ mice, $p = 0.98$; insulin, control, $n = 16$ mice, stimulation, $n = 12$ mice, $p = 0.12$; corticosterone, control, $n = 12$ mice, stimulation, $n = 5$ mice, $p = 0.24$). All data are shown as mean ± SEM. Source data are provided as a Source Data file.

activity following i.p. administration of insulin (2 U/kg[13]) using in vivo fiber photometry. We expressed a calcium sensor GCamp6s in ARC POMC neurons by injecting AAV5-Syn-FLEX-GCamp6s into the ARC of POMC-Cre mice (Fig. 6a). Although i. p. saline injection did not alter calcium signals (Fig. 6b, c), i.p. injection of insulin initially reduced POMC neuron activity, consistent with insulin-mediated inhibition of POMC neurons[3]. In fact, half of ARC POMC neurons are inhibited, while only 10% of ARC POMC neurons are excited by insulin[3]. This insulin's inhibitory effect appears to be reversed by insulin-mediated hypoglycemia as the lowest glucose levels were observed 1 h post i.p. insulin injection in our preparations. As insulin-mediated hypoglycemia increases c-fos mRNA expression in ARC POMC neurons[13], the biphasic response would be attributed to the initial insulin-mediated inhibition followed by the hypoglycemia-mediated excitation of ARC POMC neurons. Hence, consistent with the prior study[13], our results suggest that ARC POMC neurons respond to hypoglycemia with an increased neural activity.

## Discussion

It has been shown that the central melanocortin system independently regulates glucose homeostasis in addition to its regulation of energy intake and energy expenditure[5,9,12–14]. For example, central injection of the MC4R agonist reduces insulin secretion and glucose tolerance[14]. Mice lacking the *Pomc* gene exclusively in the ARC show improved glucose tolerance via elevated glycosuria[5,12]. The same mouse model also has an impaired counterregulatory response to hypoglycemia[13]. Interestingly, all these effects appear to be mediated through the sympathetic nervous system without directly altering hepatic gluconeogenesis. In this study, we provide physiological evidence that ARC POMC neurons are able to control glycemia through parasympathetic cholinergic neurons in the DMV. Neuronal mapping studies revealed that ARC POMC neurons send direct projections to DMV cholinergic neurons, and that a subset of ARC POMC neurons innervate the liver via these cholinergic neurons. Consistent with prior findings[21], activation of MC4Rs in

these neurons reduced their neuronal activity. More importantly, inhibition of liver-projecting cholinergic neurons rapidly and robustly elevated blood glucose levels. As insulin-induced hypoglycemia elevated ARC POMC neuronal activity, this ARC-POMC → DMV$^{ACh}$ → liver circuit that we identified would plays a critical role in the counterregulatory response to hypoglycemia.

ARC POMC neurons are neurochemically and neuroanatomically heterogeneous[20,33,34], suggesting the possible existence of functionally distinct POMC neuronal subpopulations. For instance, activation of β-endorphin or α-melanocyte-stimulating hormone (α-MSH)-containing POMC neurons oppositely regulate food intake[26,35]. Leptin receptor-expressing POMC neurons normalize blood glucose levels by improving hepatic insulin sensitivity without changing food intake[9]. Insulin receptor-expressing POMC neurons contribute to the control of locomotor activity and energy expenditure[36], whereas 5-HT$_{2C}$ receptor-expressing POMC neurons are involved in insulin resistance[37]. The cellular mechanisms underlying these distinct physiological effects of ARC POMC neurons have not been well explored, in part due to the lack of appropriate experimental approaches permitting selective stimulation of neuroanatomically identified POMC neuronal subpopulations in the ARC. To circumvent these technical limitations, we directly delivered the Cre recombinase transgene from peripheral organs to the ARC with AAV vectors carrying the WGA-Cre fusion protein and expressed ChR2 exclusively in ARC POMC neurons innervating the liver in a Cre-dependent manner. In contrast to the prior findings that ARC POMC neurons enhance the ability of insulin to suppress hepatic gluconeogenesis[2,3,9,37], we found that optogenetic stimulation of the ARC$^{POMC}$ → DMV$^{ACh}$ → liver projection increases blood glucose levels.

At first glance, our present results are somewhat unexpected and appear to be inconsistent with the prior findings that genetic manipulation of serotonin, insulin, leptin, and glucagon-like peptide-2 receptor signaling in ARC POMC neurons significantly improve hepatic insulin sensitivity, thereby normalizing blood glucose[2,3,9,37]. For instance, mice lacking T-cell protein tyrosine phosphatase in POMC neurons enhance insulin signaling and the ability of POMC neurons to inhibit HGP[3]. Activation of GLP-2

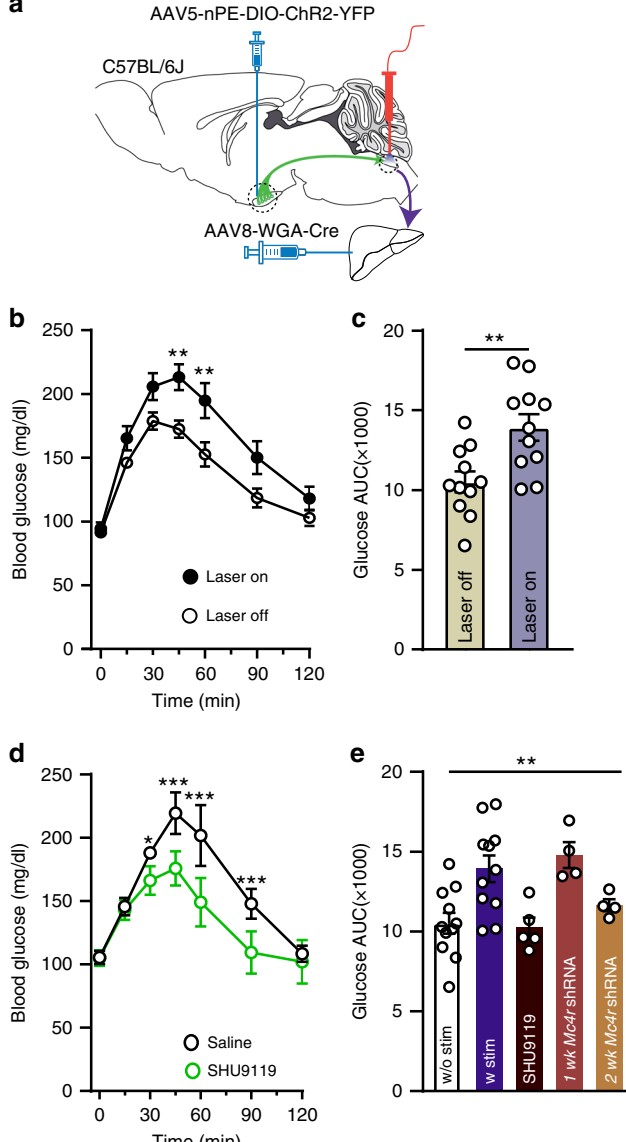

**Fig. 4 Optogenetic stimulation of the ARC$^{POMC}$ → DMV$^{ACh}$ → liver projection promotes hepatic glucose output. a** Schematic illustration of the experimental configuration. **b** Pooled data showing PTT in mice with and without optogenetic stimulation (control (open circle), $n = 11$ mice, stimulation (filled circle), $n = 11$ mice, two-way RM ANOVA followed by Sidak multiple comparison test, interaction, $F_{(6, 120)} = 1.7$, $p = 0.14$, time, $F_{(6, 120)} = 51.4$, $p < 0.0001$, between the groups, $F_{(1, 20)} = 12.6$, $p < 0.01$; 45 min, $**p = 0.01$, 60 min, $**p = 0.006$). Mice were fasted for 15 h. All data are shown as mean ± SEM. Source data are provided as a Source Data file. **c** Bar graphs showing areas under the curves (AUC) values obtained from PTT experiments (two-tailed $t$ test, $**p = 0.004$, vs. control, $n = 11$ mice). All data are shown as mean ± SEM. Source data are provided as a Source Data file. **d** Summary plot showing involvement of MC4Rs in the responses to optogenetic stimulation (saline, $n = 5$ mice, SHU9119, $n = 5$ mice, two-way RM ANOVA followed by Sidak multiple comparison test, interaction $F_{(6, 24)} = 10.7$, $p < 0.001$, time $F_{(6, 24)} = 93.0$, $p < 0.001$, between the groups, $F_{(1, 4)} = 46.0$, $p = 0.003$; 30 min, $*p = 0.02$, 45, 60 and 90 min, $***p < 0.001$). Mice were fasted for 15 h. All data are shown as mean ± SEM. Source data are provided as a Source Data file. **e** Bar graphs showing AUC values obtained from PTT experiments. Knockdown of the *Mc4r* gene was achieved by injection of *Mc4r* shRNAs into the DVC of WT mice injected with AAV8-WGA-Cre to the liver and AAV5-nPE-DIO-ChR2-YFP to the ARC (w/o stim, $n = 11$ mice, w stim, $n = 11$ mice, w stim + SHU9119, $n = 5$ mice, w stim + *Mc4r* shRNA (1 and 2 weeks), $n = 4$ mice, one-way ANOVA followed by Dunnett's multiple comparison test, between the groups, $F_{(4, 30)} = 5.9$, $p = 0.001$; without (w/o) stim vs. with (w) stim, $p = 0.003$; w/o stim vs. SHU9119, $p = 0.99$; w/o stim vs. 1-week *Mc4r* shRNA, $p = 0.008$; w/o stim vs. 2-week *Mc4R shRNA*, $p = 0.82$). All data are shown as mean ± SEM. Source data are provided as a Source Data file.

receptors in POMC neurons improves the effect of insulin on HGP[2]. Re-expression of 5-HT$_{2C}$ and leptin receptors in POMC neurons also ameliorates hepatic insulin resistance[9,37]. Moreover, chemogenetic activation of the whole population of ARC POMC neurons lowers blood glucose levels that is associated with reduced *Pepck* and *G6pase* expression in the liver, whereas chemogenetic inhibition induces the opposite effects on blood glucose and gluconeogenic enzyme expression[3]. In contrast to these findings, it has been also shown that acute chemogenetic activation of ARC POMC neurons does not influence glucose metabolism[38], while long-term inhibition of ARC POMC neurons lowers blood glucose[17]. This discrepancy may be due in part to heterogeneity of ARC POMC neurons in the context of the neurotransmitters, the receptors for hormones, nutrients, and neuropeptides, the neurophysiological responses, and importantly their distinct projection sites[2,3,8,39]. Due to this POMC neuron heterogeneity, it is difficult to interpret the findings from POMC transgenic and mutant mice. For example, acute stimulation of the whole population of ARC POMC neurons is not able to alter food intake[40,41], despite the well-known anorectic effect of α-MSH. Surprisingly, POMC-specific insulin receptor knock-in mice show exacerbated insulin resistance and increase HGP[36].

Moreover, despite the severe obesity in mice lacking POMC exclusively in the ARC, these mice display fasting euglycemia[5]. These prior findings suggest that a subset of POMC neurons may have the ability to promote hepatic glucose production. In this study, we selectively stimulated the ARC$^{POMC}$ → DMV$^{ACh}$ → liver projection among other POMC projections and this stimulation elevated blood glucose levels. Hence, our results may explain in part the cellular mechanisms underlying improved glucose tolerance and normal fasting glycemia in mice lacking the *Pomc* gene exclusively in the ARC[5] and the ability of the MC4R agonist to increase blood glucose levels via diverse cellular mechanisms[14,15,42].

In the hypothalamus, the PVN plays a key role in sending sympathetic signals to the liver[43]. For instance, infusion of the GABA$_A$ receptor antagonist and the NMDA receptor agonist into the PVN causes hyperglycemia that is blocked by a sympathetic denervation of the liver[43]. As POMC neurons project to the PVN[44] and MC4R is expressed in the PVN[45], the ARC$^{POMC}$ → PVN projection would regulate hepatic glucose production. However, the prior study suggests that MC4Rs in the PVN mainly regulate feeding[46]. In fact, injection of the MTII into the PVN reduces food intake[47] and deletion or restoration of MC4R on single-minded 1 (SIM1) neurons in the PVN controls energy expenditure, body weight, and food intake[45,48,49]. Hence, the ARC$^{POMC}$ → PVN projection appears to play a minor role in the control of hepatic glucose metabolism. Rather, MC4R-expressing PVN neurons largely contribute to the control of body weight, energy expenditure, and energy intake, whereas other types of PVN neurons regulate hepatic glucose production[50]. In fact, it has been clearly demonstrated that a subset of parvocellular neurons in the PVN are retrogradely identified liver-projection neurons[18] and that these neurons project to the intermediolateral cell column of the spinal cord[51]. Additionally, other brain regions, including the dorsomedial

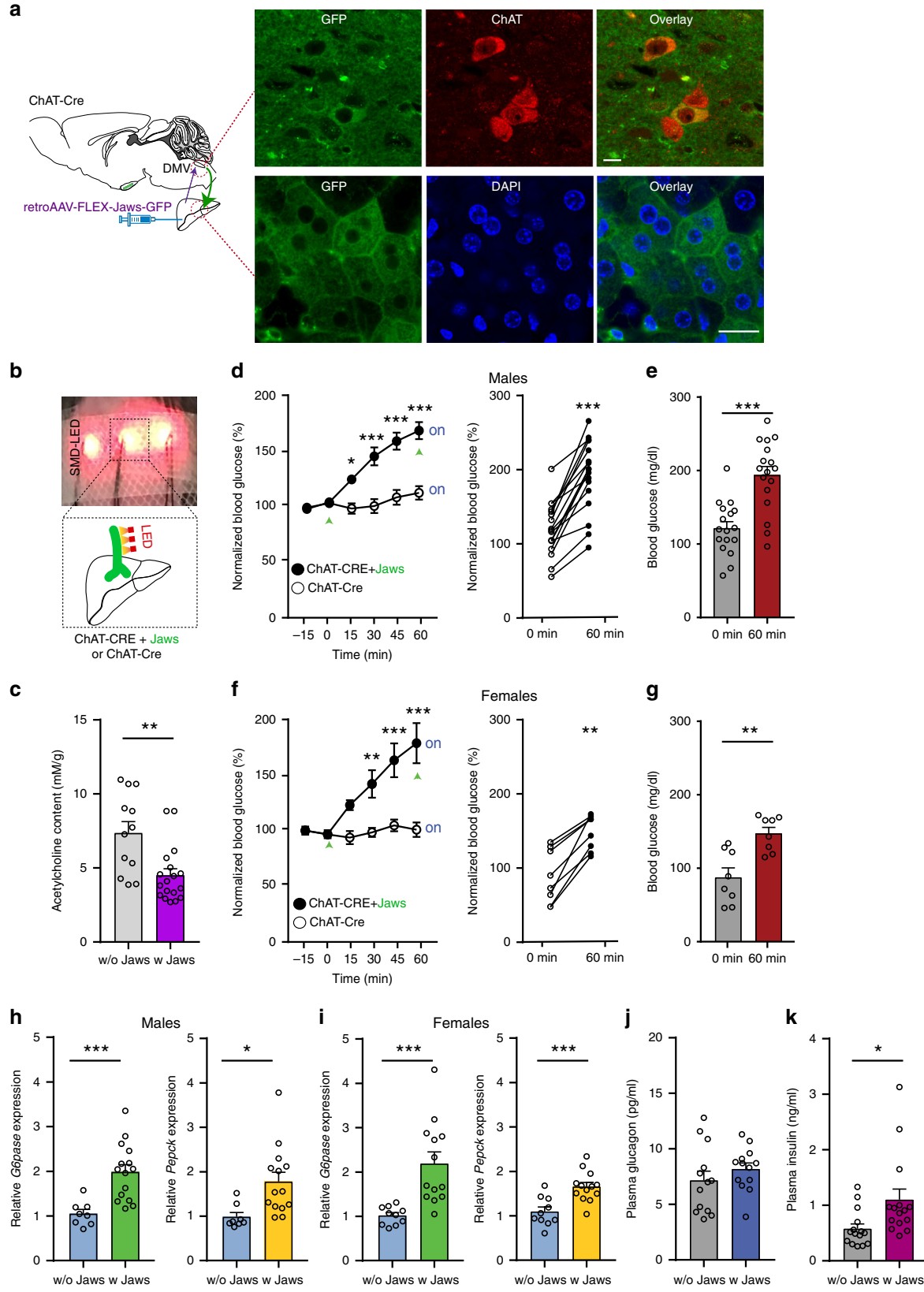

hypothalamus, ventromedial hypothalamus, central amygdala, and medulla[18], would also play a role in the control of hepatic glucose output via the autonomic nervous system.

Outside the hypothalamus, MC4R is expressed in the brainstem, including the DMV[45]. The DMV contains preganglionic parasympathetic cholinergic neurons innervating liver[22]. The role of MC4Rs in the autonomic nervous system has been documented[6]. Restoration of MC4R in autonomic cholinergic neurons (i.e. preganglionic parasympathetic and sympathetic cholinergic neurons) lowers body weight without altering food intake and

**Fig. 5 Inhibition of parasympathetic cholinergic neurons elevates blood glucose levels. a** Schematic diagram of the experimental configuration (left panel). RetroAAV2-FLEX-Jaws-GFP viruses were injected into the liver of ChAT-IRES-Cre mice. Right top panel: Images of confocal fluorescence microscopy showing co-expression of GFP (left) and ChAT (middle) in a subset of cholinergic neurons in the DMV (right panel). Scale bar: 10 μm. Right bottom panel: Images of confocal fluorescence microscopy showing the expression of GFP-positive fibers (left) around hepatocytes in the liver. Note that these are not cytoplasmic staining. Scale bar: 20 μm. **b** Pictures showing our noninvasive optogenetic stimulation. Small SMD-LED (2 mm × 2 mm) were directly placed on the skin of the abdomen following hair removal. **c** Pooled data from 12 (w/o Jaws) and 18 (w Jaws) mice showing changes in ACh content in the liver (two-tailed $t$ test, **$p = 0.002$). All data are shown as mean ± SEM. Source data are provided as a Source Data file. **d**, **e** Pooled data showing the effect of inhibition of liver-projecting cholinergic nerves on hepatic glucose output in male mice. Optogenetic inhibition of cholinergic fibers in the liver of ChAT-IRES-Cre injected with retroAAV-FLEX-Jaws-GFP mice elevated blood glucose levels (control (ChAT-IRES-Cre), $n = 8$ mice, ChAT-IRES-Cre+Jaws, $n = 17$ mice; two-way RM ANOVA followed by Sidak multiple comparisons test, interaction $F_{(5, 115)} = 13.9$, $p < 0.001$, time $F_{(5, 115)} = 24.6$, $p < 0.001$, between the groups, $F_{(1, 23)} = 19.8$, $p < 0.001$; 15 min, *$p = 0.03$; 30 min, ***$p < 0.001$; 45 min, ***$p < 0.001$; 60 min, ***$p < 0.001$; left panel). Right panel: Summary plots showing individual data before and after optogenetic stimulation (two-tailed $t$ test, ***$p < 0.001$, vs. 0 min, $n = 17$ mice) (**d**). **e** Summary plot showing changes in blood glucose levels at the end of stimulation (mean blood glucose, 0 min, 120.8 ± 8.6 mg/dl, 60 min, 192.1 ± 11.5 mg/dl, two-tailed $t$ test, ***$p < 0.001$, $n = 17$ mice). Mice were not fasted for these experiments. All data are shown as mean ± SEM. Source data are provided as a Source Data file. **f**, **g** Pooled data showing the effect of inhibition of liver-projecting cholinergic nerves on hepatic glucose output in female mice (control (ChAT-IRES-Cre), $n = 7$ mice, ChAT-IRES-Cre+Jaws, $n = 8$ mice); two-way RM ANOVA followed by Sidak multiple comparisons test, interaction $F_{(5, 65)} = 9.9$, $p < 0.001$, time $F_{(5, 65)} = 12.8$, $p < 0.001$, between the groups $F_{(1, 13)} = 14.8$, $p = 0.002$; 30 min, **$p = 0.007$; 45 min, ***$p < 0.001$; 60 min, ***$p < 0.001$; ChAT-Cre, $n = 7$ mice, ChAT-IRES-Cre+Jaws, $n = 8$ mice (**f**). **g** Summary plot showing changes in blood glucose levels at the end of stimulation (mean blood glucose, 0 min, 87.6 ± 12.8 mg/dl, 60 min, 147.1 ± 8.4 mg/dl, two-tailed $t$ test, **$p = 0.002$, $n = 8$ mice). Mice were not fasted for these experiments. All data are shown as mean ± SEM. Source data are provided as a Source Data file. **h**, **i** Bar graphs showing changes in *G6pase* and *Pepck* mRNA expression with and without Jaws following optogenetic stimulation in male (**h**) and female (**i**) mice (males, two-tailed $t$ test, *G6pase*, ***$p < 0.001$; *Pepck*, *$p = 0.01$; ChAT-Cre, $n = 8$ mice, ChAT-IRES-Cre + Jaws, $n = 14$–15 mice; females, two-tailed $t$ test, *G6pase*, ***$p < 0.001$; *Pepck*, ***$p < 0.001$; ChAT-Cre, $n = 10$ mice, ChAT-IRES-Cre+Jaws, $n = 13$ mice). All data are shown as mean ± SEM. Source data are provided as a Source Data file. **j**, **k** Bar graphs showing plasma glucagon (**j**) and insulin (**k**) levels following the inhibition of liver-projecting cholinergic nerves (two-tailed $t$ test, glucagon, $p = 0.3$ ChAT-IRES-Cre, $n = 13$ mice, ChAT-IRES-Cre + Jaws, $n = 13$ mice; insulin, ChAT-IRES-Cre, $n = 15$ mice, ChAT-IRES-Cre + Jaws, $n = 15$ mice, two-tailed $t$ test, *$p = 0.03$). All data are shown as mean ± SEM. Source data are provided as a Source Data file.

reduces blood glucose concentrations and plasma insulin concentrations[6]. In addition, MC4R re-expression in both cholinergic preganglionic sympathetic and parasympathetic neurons almost completely normalizes insulin-induced suppression in HGP[6]. In contrast, re-expression of MC4R specifically in parasympathetic neurons in Phox2c-Cre;loxTB MC4R mice does not change body weight, food intake, and energy expenditure, comparing to loxTB MC4R mice[6]. Moreover, there is no improvement of hyperglycemia that is observed in obese loxTB MC4R mice[6], suggesting that MC4Rs in DMV cholinergic neurons do not mediate the effect of melanocortins on endogenous glucose production. However, these results appear to be inconsistent with the ability of parasympathetic efferent outflow to the liver to suppress hepatic glucose output[28,29,31]. Interestingly, contrary to the previous studies describing the stimulatory effect of MC4Rs in other areas[21,52], activation of MC4Rs in the DMV inhibits rather than excites cholinergic neurons through activation of $K_{ATP}$ channels[21]. In our study, selective inhibition of liver-projecting DMV cholinergic fibers indeed elevated blood glucose levels independently of pancreatic hormones such as glucagon and the sympathetic adrenal hormone corticosterone. Importantly, our results demonstrated that hepatocytes receive cholinergic input from preganglionic parasympathetic cholinergic neurons and express mAChRs. In contrast to the traditional view that the activation of sympathetic nerves promotes[53], while the parasympathetic innervation suppresses[53], hepatic glucose production, our results provide physiological evidence that the parasympathetic nervous system alone is able to finely tune blood glucose levels through hepatic glucose metabolism in a neuronal activity-dependent manner.

What are the physiological roles of this POMC neuron-mediated increase in blood glucose levels? Hypothalamic melanocortin circuits have been shown to regulate the sympathetic nervous system[5,12,13,54]. The projection from ARC POMC neurons to PVN MC4R-expressing neurons appears to be critical for counterregulatory responses to hypoglycemia[13]. Of particular interest is that insulin-induced hypoglycemia increases ARC POMC neuron activity in vivo[13]. In line with this finding, our fiber photometry results further showed that ARC POMC neurons indeed respond to hypoglycemia with an increased neuronal activity. In this case, one can expect that hypoglycemia-mediated excitation of ARC POMC neurons will release melanocortins in the DMV that is an important downstream target of ARC POMC neurons. Furthermore, MC4R-mediated inhibition of parasympathetic cholinergic neurons will result in increased blood glucose levels. Thus, it is possible that ARC POMC neuron-mediated activation of sympathetic outflow as well as inhibition of parasympathetic cholinergic output would synergistically counteract hypoglycemia. In other words, MC4R-mediated neural interactions between parasympathetic and sympathetic nervous systems may control hepatic glucose output. Our results provide neurophysiological evidence for the link between the liver and ARC POMC neurons. However, much remains to be done to elucidate the exact cellular mechanisms, in which liver increases blood glucose levels. In fact, increased blood glucose levels in our preparations can be due to diverse cellular mechanisms, including increased hepatic glucose production, reduced glucose uptake[55], enhanced glycogenolysis or all of them. As individuals treated for diabetes are at high risk for potentially life-threatening hypoglycemia that has emerged as a leading complication of diabetes, our findings will help develop better strategies to improve counterregulatory responses to hypoglycemia in patients treated for diabetes. In addition, our experimental approaches provide powerful tools with which to study the roles of communication between the brain and peripheral tissues/organs involved in energy metabolism.

## Methods

**Ethics statement.** All mouse care and experimental procedures were approved by the Institutional Animal Care Research Advisory Committee of the Albert Einstein College of Medicine and were performed in accordance with the guidelines described in the NIH guide for the care and use of laboratory animals. Stereotaxic surgery and viral injections were performed under isoflurane anesthesia.

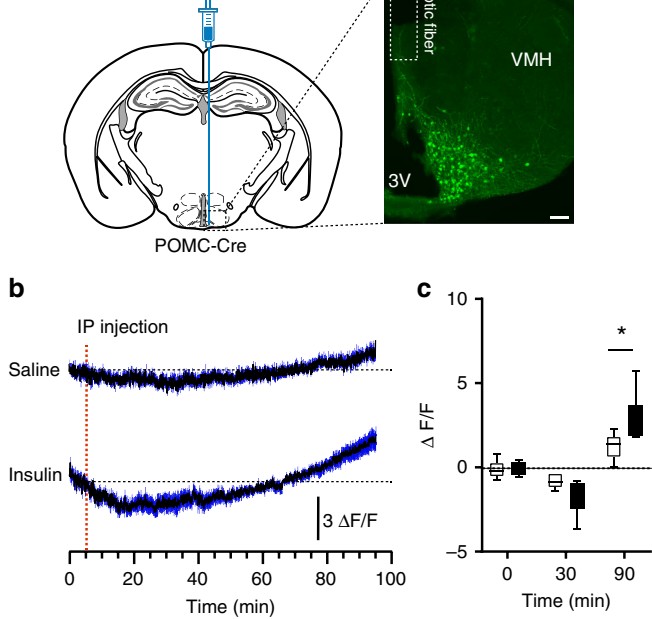

**Fig. 6 Insulin-induced hypoglycemia increases calcium signal in ARC POMC neurons. a** Schematic illustration of the experimental configuration. We injected AAV5-Syn-FLEX-GCamp6s viruses into the ARC of POMC-Cre mice (left panel). Right panel shows the expression of GFP in ARC POMC neurons in mice injected with these viruses into the ARC. Scale bar: 100 μm, 3V, third ventricle. **b** Average of normalized ARC POMC neuron calcium signal in mice treated with saline and insulin (2 U/kg) (n = 6 mice, respectively). Insulin-induced hypoglycemia caused an increase in calcium signals approximately 1 h post i.p. insulin injection. Dark blue lines represent error bars. Source data are provided as a Source Data file. **c** Summary plot showing changes in normalized ARC POMC neurons calcium signal over time in mice treated with saline and insulin (significance assessed using a two-way ANOVA followed by Sidak multiple comparisons test, 90 min, **p = 0.006). All data are shown as mean ± SEM. Source data are provided as a Source Data file.

**Animals**. Mice used in this study included C57BL/6J, POMC-Cre (stock# 005965), ChAT-IRES-Cre (stock# 006410), and floxed-stop Rosa26-eGFP mice (stock# 004077, The Jackson Laboratory). Both female and male mice of mixed C57BL/6J, FVB and 129 strain backgrounds were used. Animals were housed in groups in cages under conditions of controlled temperature (22 °C) and humidity (40–60%) with a 12:12 h light–dark cycle. Mice were fed a standard chow diet with ad libitum access to water.

**In vitro brain slice preparation and electrophysiological recordings**. We used ChAT-IRES-Cre; Rosa26-eGFP mice. To prepare transverse brain slices[26], brain was removed and placed in a cold sucrose-based cutting solution. 200 μm coronal brain slices were prepared for recordings using a vibratome (Leica Biosystems). The slices were transferred into a holding chamber, incubated in the oxygenated artificial cerebrospinal fluid (aCSF containing 5 mM glucose) at 34 °C for 1 h. For electrophysiological recordings, the brain slice was placed on the stage of an upright fluorescence BX51 microscope (Olympus life science). GFP-positive neurons in the DMV were visualized with a ×40 water-immersion objective by infrared microscopy. All recordings were made at 30 ± 2 °C. The slices were continuously superfused with aCSF at a rate of 1.5 ml/min containing the following (in mM): 113 NaCl, 3 KCl, 1 NaH$_2$PO$_4$, 26 NaHCO$_3$, 2.5 CaCl$_2$, 1 MgCl$_2$, and 5 glucose in 95% O$_2$/5% CO$_2$.

The internal solution contained the following (in mM): 130 Kgluconate, 5 CaCl$_2$, 10 Ethyleneglycol-*bis*(β-aminoethyl)-N,N,N′,N′-tetraacetic acid (EGTA), 10 4-(2-Hydroxyethyl)piperazine-1-ethanesulfonic acid (HEPES), 2 MgATP, 0.5 Na$_2$GTP, and 5 phosphocreatine. The pipette resistance was 3–6 MΩ. Membrane potentials were recorded with a Multiclamp 700B. Electrophysiological signals were low-pass filtered at 2–5 kHz, stored on a PC and analyzed offline with pClamp 10 software (Molecular Devices). For each recording, the membrane potentials and action potentials measured from every 60 s were taken as 1 data point. A total of 5

data points before and during application of drugs was compared using the two-tailed *t* test.

**Stereotaxic surgery and viral injections**. pAAV-EF1α-IRES-WGA-Cre viruses were packaged into AAV serotype 8 (titer, 3.9 × 10$^{12}$ GC/ml, Vigene). Six-week-old mice were anaesthetized deeply with 3% isoflurane. A deep level of anesthesia was maintained throughout surgical procedure. The abdominal area was cleaned with 70% ethanol prior to performing a laparotomy. Under isoflurane anesthesia (2%), these viral vectors were injected directly into the left and right lobes (2 μl per site) of the liver of C57BL/6J mice. A Hamilton syringe pump (CMA 400) was used to inject over 20 min (i.e. 100 nl per min). The needle (30G) was left for an additional 10 min to allow diffusion of the virus. The animal was allowed to recover for 14 days. At 4–6 weeks post viral injections, AAV5-nPE-DIO-ChR2-YFP[26] (200 nl of 3 × 1013 pfu/ml per side, Vigene) virus was bilaterally injected into the ARC (AP, −1.7 mm; ML, ±0.1; DV, −5.8 mm).

To optogenetically inhibit parasympathetic cholinergic nerves, retroAAV-CAG-FLEX-Jaws-KGC-GFP-ER2 viral vectors (titer, 1.3 × 1013 pfu/ml, Addgene) were injected with into the left and right lobes (2 μl per site) of the liver of ChAT-IRES-Cre mice. For in vivo fiber photometry, AAV5-Syn-FLEX-GCaMP6s-WPRE-SV40 virus (AAV5-Syn-FLEX-GCaMP6s-WPRE-SV40, titer, 1.3 × 1013 pfu/ml, UPenn vector core or AAV5-EF1α-DIO-GCaMP6s, titer, 4.1 × 1012 pfu/ml, UNC vector core) was bilaterally injected into the ARC (AP, −1.7 mm; ML, ± 0.1 mm; DV, −5.8 mm). At 3 weeks post viral injections, a mono fiber-optic cannula (length, 5.8 mm, core diameter, 400 μm, Doric Lenses) was implanted into the ARC (AP, −1.7 mm; ML, 0 mm; DV, −5.8 mm) and sealed with dental cement.

To knock down the *Mc4r* gene in the DMV, *Mc4R* (450 nl of 1.6 × 107 VP/ml, NM_016977, Sigma-Aldrich) and control shRNAs were injected into the DVC.

**Optogenetic stimulation of liver-projecting neural circuits**. Blood was collected from mouse tail every 15 min and glucose levels were measured at time −30 min (i.e. 30 min prior optogenetic stimulation) using a Bayer Contour blood glucose meter. Experiments started at 9–9:30 a.m. Mice were not fasted for these experiments. To stimulate the ARC$^{POMC}$ → DMV$^{ACh}$ → liver projection, an optic cannula was implanted into the DMV (AP, −7.5 mm; ML, 0 mm; DV, −3.5 mm) and coupled to a 473 nm DPSS laser (Laserglow Technologies). ChR2-expressing liver-projecting POMC fibers in the DMV were illuminated at 20 Hz (25 ms pulse duration, 20 Hz pulses for 1 s, 3 s interval between events, 1 h duration). Stimulation at 10–20 Hz induced a high-fidelity firing in our prior studies[26,32,56,57].

Hepatic branch vagotomy was performed in WT mice[28]. Blood glucose levels in hepatic-vagotomized mice were measured without fasting. SHU9119 (100 μg/kg) or saline was intraperitoneally injected 30 min prior to optogenetic stimulation.

With the noninvasive optogenetic stimulation technology permitting transcutaneous excitation of autonomic fibers innervating peripheral organs[56], liver-projecting parasympathetic cholinergic nerves were illuminated for 1 h under isoflurane (1.5%) anesthesia. Following anesthesia, the hair in the abdomen was removed with hair remover lotion and small (2 mm × 2 mm) surface-mounted device (SMD) LED modules (pre-soldered micro pico litz wired SMD LED, Amazon.com) were placed directly on the skin. The rationale for this method is that light penetrates through the skin of the animals. Each SMD-LED module has low voltage requirement and was connected to an optogenetics TTL pulse generator (OPTG_4, Doric lenses). This TTL pulse generator was connected to a PC computer. Bursts of light pulses were generated with the aid of computer software (Doric Neuroscience Studio, Doric lenses). Light was directly applied through the skin. A warm pad was used to maintain the animal's body temperature and the body temperature was continuously monitored with a THERMES-USB temperature data acquisition system (Physitemp).

**Pyruvate tolerance test and glucose tolerance test**. C57BL/6J mice injected with AAV8-WGA-Cre into the liver and AAV5-nPE-DIO-ChR2-YFP into the ARC were fasted for 15 h (18:00 p.m.−9:00 a.m.) and received an i.p. injection of pyruvate (2 g/kg body weight) and blood samples were collected from mouse tail at 0, 15, 30, 45, 60, 90, and 120 min to measure circulating glucose concentrations without optogenetic stimulation. Another set of animals was conducted with optogenetic stimulation. Mice received an i.p. injection of saline or SHU9119 (100 μg/kg) 15 min prior to optogenetic stimulation. Blood glucose levels vs. time after pyruvate injection was plotted, and area-under-curve was calculated.

For glucose tolerance test (GTT), mice were fasted for 15 h (18:00 p.m. −9:00 a.m.) before being subjected to glucose tolerance test. For GTT, mice were intraperitoneally injected with glucose (2 g/kg body weight, Sigma-Aldrich). Both tests were conducted in awake mice.

**Quantitative real-time PCR analysis**. For qPCR analysis of *G6pase Pepck, and Mc4r* genes, total RNAs were isolated using the RNeasy mini kit (Qiagen, 74104) from liver and DMV tissues and then first-strand cDNAs were synthesized using the SuperScript III First-Strand synthesis kit (Thermo Fisher Scientific, 18080-051). Real-time qPCR was performed in sealed 96-well plates with SYBR Green I master Mix (Applied Biosystems, A25742) using a Quant Studio 3 (Applied Biosystems). qPCR reactions were prepared in a final volume of 20 μl containing 2 μl cDNAs and 10 μl of SYBR Green master mix in the presence of primers at 0.5 μM. β-actin

was used as an internal control for quantification of each sample. qPCR analysis of *Chrm1* to 5 genes in liver was also performed. Supplementary Table 1 contains a complete list of all primers used in this study. Amplification was performed under the following conditions: denaturation at 95 °C for 30 s, followed by 40 cycles of denaturation at 95 °C for 30 s and annealing/extension at 60 °C for 1 min. The relative expression levels were determined using the comparative threshold cycle (CT), which was normalized against the CT of β-actin using the $^{\Delta\Delta}Ct$ method.

**Immunofluorescence staining**. Mice were anesthetized with isoflurane (3%) and transcardially perfused with pre-perfusion solution (9 g NaCl, 5 g sodium nitrate, 10,000 U heparin in 1 L distilled water). Brains and livers were incubated in 4% paraformaldehyde overnight at cold room and sectioned with a vibratome in 40 μm on the following day. The sections were blocked in 0.1 M PBS buffer containing 0.2 M glycine, 0.1% triton X-100, 10% normal goat serum or normal donkey serum, and 5% bovine serum albumin for 2 h at room temperature and then incubated with mouse anti-GFP (1:1000, Invitrogen, cat# A-11120), mouse anti-YFP (1:1000, Clontech, cat# 632380), rabbit anti-POMC (1:1000, Phoenix pharmaceuticals, cat# H-029-30), and goat anti-ChAT (1:100, Millipore, cat# AB144-P) antibodies for 72 h at cold room, and then sections were washed three times in PBS and incubated with Alexa 488 anti-rabbit IgG (1:500; Life Technologies, cat# A21206), Alexa 488 anti-mouse IgG (1:500; Life Technologies, cat# A21202), Alexa 568 anti-rabbit IgG (1:500; Life Technologies, cat# A10042), Alexa 568 anti-goat IgG (1:200; Life Technologies, cat# A11057), Alexa 568 anti-mouse IgG (1:500; Life Technologies, cat# A11004) for 2 h at room temperature. Tissues were washed, dried and mounted with VECTASHIELD media containing DAPI. Images were acquired using a Leica SP5 confocal microscope. Cell counting was carried out with ImageJ software (version Fiji).

**Measurement of plasma insulin, glucagon, corticosterone, L-lactate, and epinephrine**. Blood samples were collected from the retroorbital plexus with heparinized capillary tubes (VWR international, LLC) with and without 1-h optogenetic stimulation and then centrifuged at $15,600 \times g$ for 10 min to collect plasma. Plasma insulin and glucagon levels were quantified using the ELISA kits (Mercodia, 10-1281-01 for glucagon, and 10-1247-01 for insulin, respectively) according to protocol provided by Mercodia. Optical density was measured at 450 nm using a microreader. Plasma corticosterone levels were measured using an ELISA kit (Enzo Life Sciences, ADI-900-097) according to the manufacturers' instructions.

For measurement of plasma L-lactate and epinephrine, blood samples were collected from ChAT-IRES-Cre mice injected with retroAAV-FLEX-Jaws and ChAT-IRES-Cre mice following 1-h light illumination. The collected blood samples were centrifuged at $15,600 \times g$ for 10 min at 4 °C. Plasma L-lactate and epinephrine levels were quantified using the colorimetric assay kits (Cayman, 700510 for L-Lactate, and Novus Biologicals, NBP2-62867 for epinephrine, respectively).

**Measurement of hepatic acetylcholine content**. Liver tissues were collected from ChAT-IRES-Cre mice injected with retroAAV-FLEX-Jaws and ChAT-IRES-Cre mice following 1 h of light illumination. Fifty-five milligrams of liver tissue were homogenized in chloroform/methanol (2:1, v/v). After centrifugation at 4 °C, the homogenate was incubated at room temperature for 1 h on an orbital shaker and phase separation was induced by adding distilled water. The lower organic phase was collected. The upper phase of the sample was re-extracted with chloroform/methanol/water (86:14:1, v/v/v). Organic phases were combined and dried in a vacuum centrifuge. Then, samples were dissolved in chloroform/methanol/water (60:30:4.5, v/v/v). Acetylcholine levels were quantified using the Acetylcholine assay kit (Cell Biolabs, Inc., STA-603).

**In vivo fiber photometry recordings**. We performed the fiber photometry in awake mice using a one-site Fiber Photometry System (Doric lenses). We measured the 405 nm-excited calcium-independent GCaMP fluorescence as a control and the 465 nm-excited calcium-dependent GCaMP fluorescence, providing a measure of POMC neuron activity with lock-in amplification. A mono fiber-optic cannula with 400 μm core diameter and 5.8 mm length (Doric lenses, MFC_400/430-0.57_5.8mm_MF1.25_FLT) was implanted to the ARC and connected to two LEDs (405 and 465 nm) connected to a PC computer. The combined excitation light was sent into an optical fiber patch cord made of a 400-μm core, NA 0.48, low-fluorescence optical fiber (Doric lenses, MFP_400/430/LWMJ-0.48_1m_FCM-FCM_T0.05). The patch cord was connected to an implanted fiber via a Zirconia Sleeve (Doric Lenses, SLEEVE_ZR_1.25 mm). GCaMP6 emission fluorescence signals were collected through the same patch cord. Excitation light intensities were modulated at different frequencies (208 and 327 Hz for 405 and 465 nm, respectively). Each channel was set to 250–300 mA for the 405 nm LED, and 250–400 mA for the 465 nm LED. In lock-in mode, we sampled the raw signal at 12 kHz. To reduce the file size during data acquisition, we used the Decimation option in the saving option. A decimation factor of 100 was used, providing 120 points per second. Demodulated calcium signals were recorded online with Doric Neuroscience studio software version 5.3 (Doric lenses). Baseline GCaMP6 fluorescence was recorded prior to an i.p. injection (20 min). Data were analyzed with Graphpad Prism 7.0.

To induce hypoglycemia, mice received an i.p. insulin injection (2 U/kg in PBS)[13] and baseline glycemia was recorded as 0 min measurement. Following an i.p. insulin or saline injection, blood glucose was measured at 15, 30, 45, 60, and 120 min. This offered the time course of blood glucose disposal upon insulin administration. In another set of experiments, we recorded in vivo calcium signals before and after i.p. injection of saline or insulin to avoid any disturbance while recording. To minimize handling-induced stress that may alter calcium signal, mice were habituated to daily handing and i.p. saline injections for at least 7 consecutive days.

**Statistics and reproducibility**. All statistical results are presented as mean ± SEM. Statistical analyses were performed using Graphpad Prism 7.0. Two-tailed Student's *t* tests were used to calculate *p* values of pair-wise comparisons. Data for comparisons across more than two groups were analyzed using a one-way ANOVA with post hoc comparisons. Time course comparisons between groups were analyzed using a two-way repeated-measures (RM) ANOVA with Sidak's correction for multiple comparisons. Data were considered significantly different when probability value was <0.05. Immunofluorescence staining was conducted at least seven and eight times to ensure reproducibility.

**Reporting summary**. Further information on research design is available in the Nature Research Reporting Summary linked to this article.

## Data availability
All data associated with this study are present in the paper or Supplementary information. A reporting summary for this article is also available as a Supplementary Information file. Source data are provided with this paper.

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

## Acknowledgements

We thank Dr. Rajat Singh for his valuable feedback and comments on this study. We also thank Tae-Sik Yoo for viral injections and Jae-Hoon Jeong who had initiated this study. This work was supported by the NIH (RO1 DK092246 and R03 TR003313) and the Foundation for Prader-Willi Research to Y.-H.J. and P30 DK020541 to Y.-H.J., S.C.C., and G.J.S.

## Author contributions

E.K. performed immunocytochemistry, real-time qRT-PCR, and optogenetics experiments, and analyzed the data. H.-Y.J. carried out viral injection, optogenetics, and fiber photometry experiments and analyzed data. S.-M.L. and S.C.C. designed and made plasmids. G.J.S. performed hepatic vagotomy. Y.-H.J. designed research, performed immunocytochemistry, optogenetics and fiber photometry experiments, analyzed the data and wrote the manuscript with inputs and comments from S.C.C. and G.J.S.

## Competing interests

The authors declare no competing interests.
