## [Peer Review File · Nature Communications]

Reviewers' Comments:

Reviewer #1:

Remarks to the Author:

Kwon and colleagues describe results from a series of studies investigating the physiological role a neuronal pathway comprising the melanocortin system from the arcuate nucleus of the hypothalamus, the motor parasympathetic nervous system and the liver. Using a combination of viral vectors, transgenic mice, optogenetics and physiological measurements, the researchers challenge the notion of the melanocortin system as predominantly anti diabetic and propose an additional role of the POMC neurons to increase systemic glucose levels.

Figure 3, Control optogenetics experiments using AAV with fluorescence only are missing. Currently, we cannot completely rule out the possibility that thermal effects on DMV may have influenced the results. The same issue exists throughout all results from optogenetics experiments. Also, they should validate proper expression of their optogenetics constructs using electrophysiology or FOS immunohistochemistry experiments.

Figure 5, the authors injected rAAV into the liver and placed LED on abdomen. While the authors intended to infect cholinergic neurons within the DMV, and some DMV neurons were indeed infected as shown in Figure 5a, the parasympathetic ganglions in the liver also contain cholinergic neurons. Therefore, the GFP signal observed in the hepatocytes (Figure 5a & Supplementary Figure 5a) as well as the observed ChAT signal (Supplementary Figure 5a) is likely to originate from the parasympathetic ganglionic (not preganglionic) neurons. Taken together, these results do not properly represent the effects of inhibiting DMV cholinergic neurons.

Figure 6, the magnitude of fluorescent signal increase is quite small. The difference between saline and insulin treatments is only $\sim 1\%$, while they are shown to be statistically significant. Also, the average was increased even by saline treatments. Would this small difference be sufficient to result in significant increase of alpha-MSH secretion? It may not be appropriate to suggest that this effect underlies hypoglycemia-induced hepatic gluconeogenesis unless any significant increase of alpha-MSH secretion is shown.

The researchers confirmed the existence of a neuronal pathway from the POMC neurons to the cholinergic neurons in the DMV using anterograde tracing and immunohistochemistry. A second tracing was performed to corroborate the POMC-DMV-liver pathway by injecting a virus expressing Cre recombinase fused to the transcellular tracer protein, wheat germ agglutinin (WGA) in the liver. Despite the fact that previous reports showed that WGA-Cre is preferentially anterograde and mostly monosynaptic. The researchers report successful trans synaptic retrograde tracing in mice expressing the fluorescent reporter GFP in a Cre dependent manner. Since the vagus nerve is a mixture of sensory and motor fibers, and the AAV8 doesn't appear to be selective for a specific neuronal subtypes, it is unclear if the 20% of POMC neurons reported successfully labeled after tracing innervate the liver via the DMV or are also linked to other nuclei (e.g. NTS).

Using the trans neuronal Cre trafficking from the liver in combination with a virus injected in the ARC expressing ChR2 with a Neuronal POMC enhancers (nPEs), the researchers avoided expression of ChR2 in other cells connected to POMC neurons in the ARC. They confirmed the presence of POMC fibers in the cholinergic neurons in the DMV, however, similar to the previous tracing experiment, it is unclear if these projections are exclusive of the DVM or (and most likely) POMC fibers are also present in the surrounding area (i.e. NTS, AP) this is important because the implant was situated above the brainstem, thus all de dorsal vagal complex received light stimulation.

The authors injected a Cre dependent retroAAV expressing the red-shifted, light-driven inward chloride pump Jaws to silence neuronal activity in Chat cre mice to inhibit hepatic parasympathetic

afferents with red light delivered transdermal. Jaws has been reported to cause stronger photocurrents than other opsins. It has been previously reported that an additional consequence of strong ion-pumping activity is an increased firing rate when illumination is abruptly terminated, especially when expressed at high levels. In contrast with the present findings, previous invasive and noninvasive approaches have reported lower levels of glucose after parasympathetic stimulation. Although the authors used a low frequency (1Hz), given the lack of electrophysiological recordings of DMV neurons, it is still possible that the effect observed on glucose could be a side effect of increased firing rate after inhibition.

To evaluate a physiological condition that might require activation of the POMC neurons projecting to the liver to increase glucose and counteract hypoglycemia, the researchers followed the activity of the POMC neurons after insulin administration using *in vivo* fiber photometry. In contrast to their previous experiments using the retrograde Cre system, the authors used POMC cre mice and injected a viral vector to express the calcium sensor GCAMP6s in a cre dependent manner, therefore they registered the activity of any POMC neuron, regardless of their connectivity. Typically, hypoglycemia is observed as early as 15 min after insulin administration in non-obese mice, however, POMC activity registered by fiber photometry increased only after 60 min. This delayed response is somehow inconsistent with the fast response observed by optogenetic stimulation (15 min), the authors did not discuss possible explanations for this difference in the response of POMC neurons to hypoglycemia vs the fast effects observed by optogenetic stimulation.

Minor

In introduction, the authors mention glucose-lowering effects of POMC neurons (1st paragraph) and then move on to glucose-raising effects of the melanocortin system (2nd paragraph). Here, they link both effects to the sympathetic nervous system, which is confusing.

The authors focused on hepatic gluconeogenesis. However, hepatic glucose production is also from glycogenolysis, as the authors mentioned in introduction. Is there any reason why the authors did not examine the latter?

Results, page 5, first paragraph, ChAT is for choline acetyltransferase, not choline transferase.

Figure 3h-j, labels for x axes are missing.

Figure 4e, scrambled (control) shRNA data needs to be added.

Reviewer #2:

Remarks to the Author:

General comments

This study uses sophisticated nerve tracing and optogenetic techniques to map neural innervation of the liver. The mapping experiments appear to be well done and the resulting data are of much interest. The weakness of the paper lies in the limited approach to quantifying the effects of neural input on liver glucose metabolism. To ascribe a rise in glucose to an increase in glucose production is a stretch. Direct measurement of glucose production would have been much more convincing. Likewise, to infer that changes in glucose production are related to increased transcription of G6Pa and PEPCk is probably not correct. The impact of changes in parasympathetic input to the liver are nonetheless interesting.

Specific comments

1. Please provide the duration of fast for the animals in each of the figure legends. This is important since a 15 hr fast probably emptied the liver of glycogen and this supports the authors' conclusions regarding gluconeogenesis. It is not appropriate to conclude that the plasma glucose level had plateaued at sixty min in Fig. 3. It was clearly still rising. Further, to assume the rise in plasma glucose reflects glucose production is speculative. The liver takes up a small amount of

glucose even after a fast when it is in a net production mode. In addition, in the presence of increased glucose and insulin, hepatic glucose uptake is even more likely to occur. The question then arises as to whether the increase in plasma glucose is secondary to an increase in liver glucose production, a decrease in liver glucose uptake or a combination of the two. It would be interesting to know whether glucokinase transcription changed. It would be even more interesting to know if its location in the hepatocyte changed. It seems to me that the best way forward is to conclude that the rise in glucose was a result of an effect on the liver and that could represent either an increase in glucose production or a decrease in glucose production uptake or both.

2. The optogenetic increases in mRNA for G6Pas and PEPCK at 1 hr were not large and it seems unlikely that the protein levels of these enzymes would have been altered at sixty min, let alone after the first 15 min when the initial rise in glucose occurred. Did the authors ever measure glycogen in the liver to ensure that they were depleted? Such would be expected after a 15 hr fast but it would be reassuring to know that it were actually measured.

3. Given the rate of rise in glucose, glucose production must have exceeded glucose utilization for the full hour of stimulation even though glucose utilization must have risen due to increases in both glucose and insulin. If gluconeogenesis explains the rise in glucose production, where did the substrates come from? The increase in glucose and insulin would have decreased lipolysis and reduced the flow of FFA and glycerol to the liver. Production of amino acids by muscle would not have been expected to increase leaving lactate as the only carbon source. Did the authors' measure lactate? Further, what would the site of allosteric regulation be such that it could activate quickly and produce a rise in plasma glucose in 15 min?

4. Is there any chance that plasma epinephrine was increased. This would have caused a large flux of lactate from muscle to the liver, which in turn may have increased glucose production. It would also have inhibited glucose uptake by muscle possibly explaining at least in part the ensuing hyperglycemia.

5. The intraperitoneal GTT data are interesting but the difference in plasma glucose is primarily due to a decrease in hepatic glucose uptake rather than increase in hepatic glucose production. The combination of IP glucose delivery, hyperglycemia and hyperinsulinemia can cause the liver to take up glucose at very high rates. This would suggest that normal parasympathetic input to the liver is important to glucose tolerance as suggested by Shimazu et al. many years ago. Did you measure insulin levels in the glucose tolerance experiments? Presumably, it would be higher in the laser on group making the finding even more compelling. You should also keep in mind that the level of insulin at the liver is 3 times higher than it is in peripheral blood.

6. There are numerous places in the discussion where the plasma glucose change has been attributed to glucose production. The authors should either measure glucose production or they should take a more cautious approach to interpreting the data. The value of the paper lies in the link between the liver and the ARC. The exact mechanism, in which liver is increasing glucose levels, remains to be clarified.

Reviewer #3:

Remarks to the Author:

Kwon et al. investigate the role of POMC-DMV projections in the regulation of hepatic glucose production. They report that ARCPOMC neurons provide parasympathetic input to liver via relay in Chat DMV neurons. Optogenetic stimulation of ARCPOMC projections in the DMV promotes increases in blood glucose concentrations and hepatic glucose production as assessed via PTT, while inhibition of parasympathetic cholinergic neurons also promotes increased blood glucose

concentrations. Finally, POMC neuron fiberphotometry supposedly reveals increased POMC neuron activation 90 minutes after insulin injection.

The present study provides a model according to which POMC neurons projecting to parasympathetic cholinergic neurons in the DMV inhibit their activity to promote hepatic glucose production. Although potentially interesting, the study has numerous technical and conceptual problems.

1. The YFP-positive POMC fibers in Fig. 1g, h are barely visible. Moreover, co-staining for endogenous α -MSH should be performed to characterize them as bona fide POMC neuron projections.
2. The authors take an approach of combined retrograde Cre transport from liver with a POMC enhancer and Cre dependent expression of ChR upon AAV injection in the ARC. Although this is a potentially elegant approach to specifically target liver innervating POMC neurons, it should be complemented to stimulating DMV-projecting fibers in POMC Cre mice injected with a Cre-dependent ChR-expressing virus in the ARC.
3. This is particularly important, since the present study is in striking contrast to earlier work by the Elmquist lab, showing that selective re-expression of the MC4R in Chat-expressing neurons in the DMV IMPROVES glucose metabolism and SUPPRESSES hepatic glucose production (Rossi et al., Cell Metab. 2011). The authors of the present manuscript incorrectly refer to this study, when they state on page 15 "Moreover, there is no improvement in hyperglycemia that is observed in obese loxTB MC4R mice". As shown in Fig. 5 of the study by Rossi et al., selective re-expression of MC4Rs in Chat neurons, reduces blood glucose concentrations (Fig. 5a), plasma insulin concentrations (Fig. 5b), improves GIR during a clamp (Fig. 5e) and most importantly almost completely normalizes insulin-induced suppression in HGP (Fig. 5g). There is no solution offered for this striking discrepancy.
4. The presented Ca imaging studies are not impressive. First, the technical description how they were performed and analyzed is minimal. Second, the quantitative effects are marginal, i.e. 1% difference in dF/F comparing saline and insulin at 90 minutes after insulin injection. Third, there is no rationale provided for this kinetic. Upon injection of the used dose of insulin hypoglycemia should be apparent after 30 minutes (no data are provided for this, but based on earlier studies (Tooke et al., Mol. Metab. 2019)). Strikingly, Tooke et al., report a robust induction of Fos mRNA expression in POMC neurons 30 minutes after an injection of the same dose of insulin. There is no reason, why direct assessment of Ca activity should occur only 60 minutes after this event. In fact it is predicted to occur prior to changes in Fos expression, if anything meaningful was measured.

We thank the Editor and expert Reviewers for their careful and detailed review of the manuscript. In revising the manuscript, we have addressed all concerns raised by the reviewers. Our responses and revisions to the manuscript are detailed in this rebuttal letter.

REVIEWER COMMENTS

Reviewer #1:

Re: 1-1. Figure 3, Control optogenetics experiments using AAV with fluorescence only are missing. Currently, we cannot completely rule out the possibility that thermal effects on DMV may have influenced the results. The same issue exists throughout all results from optogenetics experiments. Also, they should validate proper expression of their optogenetics constructs using electrophysiology or FOS immunohistochemistry experiments.

-. We agree, and thank the Reviewer for these constructive suggestions. Accordingly, we conducted additional experiments with mice injected with control viruses (Suppl. Fig. 1f, g, and h). We should also emphasize that the results shown in Fig. 5 and supplementary Fig. 3d and e were obtained from mice illuminated with light, further indicating that light illumination itself does not change blood glucose levels in our preparations.

Recent studies have demonstrated that high light powers (10–15 mW/mm²) cause heating and can reduce firing rate (Owen et al., Nat. Neurosci. 2019). In fact, early optogenetic studies use high light powers because they can be shown to induce spiking with high fidelity (i.e. 7.2 mW/mm² light: 96.7 ± 3.2% spike fidelity; 23.8 mW/mm² light: 99.0 ± 1.0% spike fidelity; Zhang et al., Nature Neurosci., 2008). Since then, brain tissue has typically been illuminated using high light powers (10-30 mW). As we were aware of this issue (Jeong et al., PLOS Biol., 2018), we have been using relatively low light powers (< 3mW) of LED or laser. As shown in our prior studies (Jeong et al., Mol. Metabol., 2015; Lee et al., Nature Comm. 2015), this lower light power induced spiking without any accommodation and/or adaptation.

Additionally, we performed immunostaining for pS6 (a neuronal activity marker; Knight et al., Cell, 2012; Lee et al., Nat. Commu. 2015) to validate proper expression of our constructs (Ref. Fig. 1A and B). Light illumination increased pS6 expression in ARC POMC neurons in mice injected with both AAV-WGA-Cre and AAV-nPE-DIO-ChR2-YFP, whereas light illumination did not induce pS6 expression in ARC POMC neurons in mice injected with AAV-nPE-DIO-ChR2-YFP only. We also note that we have successfully used the same constructs to stimulate TRPV1 receptor-expressing POMC neurons in the ARC (Jeong et al. PLOS Biology, 2018). This prior study further validates the expression and efficiency of our constructs.

Reference Figure 1) Images of confocal fluorescence microscopy showing expression of pS6 protein, a neuronal activity marker, in ARC POMC neurons following light illumination. **A.** C57BL/6J Mice were injected with AAV5-nPE-DIO-ChR2-YFP only. **B.** C57BL/6J Mice were injected with both AAV5-nPE-DIO-ChR2-YFP and AAV-WGA-Cre. The ARC was illuminated with light in both cases. White arrows indicate co-expression of pS6 and POMC in ARC POMC neurons.

Re: 1-2. Figure 5, the authors injected rAAV into the liver and placed LED on abdomen. While the authors intended to infect cholinergic neurons within the DMV, and some DMV neurons were indeed infected as shown in Figure 5a, the parasympathetic ganglions in the liver also contain cholinergic neurons. Therefore, the GFP signal observed in the hepatocytes (Figure 5a & Supplementary Figure 5a) as well as the observed ChAT signal (Supplementary Figure 5a) is

likely to originate from the parasympathetic ganglionic (not preganglionic) neurons. Taken together, these results do not properly represent the effects of inhibiting DMV cholinergic neurons.

- Yes, it is possible that postganglionic parasympathetic cells can regulate hepatic glucose metabolism. However, **there is a reported lack of postganglionic parasympathetic cells in the liver** (Akiyoshi et al., Liver, 1998; McCuskey, Anat. Record, 2004). In other words, **NO clear intrahepatic postganglionic neurons have been identified**, unlike the case for other visceral organs (Berthoud et al., Neurogastroenterol. Motil., 2004).

To confirm the prior findings, we conducted additional immunohistochemical experiments with anti-choline acetyltransferase (ChAT) and anti-neuronal markers such as microtubule-associated protein 2 (MAP2) and NeuN antibodies. Consistent with the prior findings, including our own (Suppl. Fig. 7A), we detected ChAT-positive puncta (i.e. cholinergic axon terminals), **but no ChAT-positive cell bodies in liver tissues**. Furthermore, there were no MAP2- or NeuN-positive cells in the liver (Ref. Fig. 2). Therefore, there are **NO** cholinergic postganglionic neurons in the liver. These results support the interpretation that inhibiting preganglionic parasympathetic cholinergic neurons that project to the liver elevates blood glucose levels.

Reference Figure 2) Images of confocal fluorescence microscopy showing no postganglionic parasympathetic cholinergic neurons in the liver. A. Double staining with anti-MAP2 and ChAT antibodies. B. Double staining with anti-NeuN and ChAT antibodies.

RE: 1-3. Figure 6, the magnitude of fluorescent signal increase is quite small. The difference between saline and insulin treatments is only ~1%, while they are shown to be statistically significant. Also, the average was increased even by saline treatments. Would this small difference be sufficient to result in significant increase of alpha-MSH secretion? It may not be appropriate to suggest that this effect underlies hypoglycemia-induced hepatic gluconeogenesis unless any significant increase of alpha-MSH secretion is shown.

- We completely agree with the Reviewer. We regraphed Fig. 6 to clarify the differences we observed. This new figure shows a clear difference between the two groups.

Indeed, the difference in calcium signal between the control and treatment groups is small. However, we should emphasize that the difference in calcium signal *in ARC POMC neurons* is generally quite small. For example, the work from the Betley lab (Alhadeff et al. Neuron, 2019) shows changes in calcium signal in ARC POMC in response to different substances (Ref. Fig. 3A and B). Changes in calcium signal in ARC POMC neurons range from 0.2% to 0.5% $\Delta F/F$. This would be due in part to the fact that this assessment reflects **the summed activity** of the entire POMC population. Multiple, accumulating studies have observed that ARC POMC neurons are heterogeneous in the context of the neurotransmitters/ neuropeptides, the receptors for hormones and nutrients, and the respective neurophysiological responses. The altered POMC neural activity represents a sum of this heterogeneous POMC neuronal activity. For instance, 10% of ARC POMC neurons are excited, while half of them are inhibited by insulin. A third of POMC neurons do not respond to insulin (Dodd et al., Elife, 2018; Qiu et al., Cell Metab., 2014). The heterogeneous neurophysiological responses of ARC POMC neurons to neurotransmitters (Sohn et al., Neuron, 2011), hormones (Lee et al., Nat. Comm., 2015), and nutrients (Patron et al., Nature, 2007) are very common. Hence, the **net** neural activity expressed as $\Delta F/F$ may in fact be expected to be small.

In our preparations, i.p. injection of insulin initially reduced POMC neuron activity, consistent with insulin-mediated inhibition of POMC neurons. In fact, half of ARC POMC neurons are inhibited, while only 10% of ARC POMC neurons are excited by insulin (Dodd et al., Elife, 2018). These inhibitory effects of insulin appear to be reversed by insulin-mediated hypoglycemia as the lowest glucose levels were observed 1 hr post i.p. insulin injection in our preparations. As insulin-mediated hypoglycemia increases c-fos mRNA expression in ARC POMC neurons (Tooke et al., Mol Metab., 2019), the biphasic response could be attributable to an initial insulin-mediated inhibition followed by a hypoglycemia-mediated excitation of ARC POMC neurons. Interestingly, **i.p. injection of glucose does not depolarize ARC POMC neurons** (Ref. Fig. 3B), which is in contrast to the neurophysiological response of ARC POMC neurons to glucose (Patron et al., Nature, 2007;

blue arrow). Accordingly, we posit that our observed calcium signal reflects the activity of ARC POMC neurons in response to insulin-induced hypoglycemia.

Reference Figure 3) A and B. Changes in calcium signal in ARC POMC neurons in response to different substances. The responses ranged from 0.2% to 0.5%. Note that i.p. injection of glucose induced **NO** effect (blue arrow). Figures were taken from the work by the Betley group (Alhadeff et al. Neuron, 2019). **C and D.** i.p. injection of insulin induced the biphasic response. Pooled data from 5 mice showing changes in blood glucose levels post i.p. injection of insulin (D).

RE: 1-4. The researchers confirmed the existence of a neuronal pathway from the POMC neurons to the cholinergic neurons in the DMV using anterograde tracing and immunohistochemistry. A second tracing was performed to corroborate the POMC-DMV-liver pathway by injecting a virus expressing Cre recombinase fused to the transcellular tracer protein, wheat germ agglutinin (WGA) in the liver. Despite the fact that previous reports showed that WGA-Cre is preferentially anterograde and mostly monosynaptic. The researchers report successful trans synaptic retrograde tracing in mice expressing the fluorescent reporter GFP in a Cre dependent manner. Since the vagus nerve is a mixture of sensory and motor fibers, and the AAV8 doesn't appear to be selective for a specific neuronal subtypes, it is unclear if the 20% of POMC neurons reported successfully labeled after tracing innervate the liver via the DMV or are also linked to other nuclei (e.g. NTS).

- We agree with the Reviewer. WGA has been used for anterograde as well as retrograde neuronal tracing. We chose to use WGA instead of other retrograde transsynaptic tracers, such as pseudorabies virus (PRV). In fact, there are a number of issues with pseudorabies virus. First, PRV is lethal. Adult mice typically survive only five to seven days after injection with the PRV strain. Therefore, PRV is not appropriate for experiments that require survival times longer than one week. Second, infected animals typically display signs of illness during the survival period, thereby limiting the utility of PRV for experiments requiring observation of normal behavior. Third, PRV triggers a cytotoxic immune response. Five days after viral injection, 3rd or 4th order neurons may be detectable, but the earliest infected neurons may become apoptotic. Perhaps most importantly and finally, the study from the Friedman group (Stanley et al., PNAS, 2010) describes that they observe labeled neurons in the hypothalamus only at 6- 7 days after hepatic PRV infection. Moreover, only 2% of POMC neurons are labeled with PRV. This very low number of labeled POMC neurons appears to be due to a very short survival time and slow transport of PRV from the liver to the ARC.

Interestingly, it looks like that WGA is preferentially transported in a retrograde direction in the peripheral nervous system (Sugita and Shiba, Science, 2005; Ohmoto et al., Mol. Cell. Neurosci., 2008; Bai et al., Cell, 2019, etc). Moreover, WGA is among the first genetically targeted transsynaptic tracers from the PNS to the CNS (Horowitz et al., PNAS, 1999). Although this tracer can cross synapses, most studies with WGA are successfully able to label second, but not third order neurons. As noted by the Reviewer, it is plausible that not only motor neurons but also sensory neurons can take up and transport AAV-WGA-Cre viruses toward their cell bodies. The liver is innervated by two different sensory neuronal populations; one in the nodose ganglion and the other in the dorsal root ganglia (DRG). In these cases, WGA should cross more than three-four synapses to reach the hypothalamus. In fact, sensory neurons in the nodose ganglion send projections to the

NTS and then the NTS send sensory vagal afferent information to diverse areas, including the parabrachial nucleus (PB), ventrolateral medulla (VLM), hypothalamus, and the bed nucleus of the stria terminalis (BNST). **Importantly, there is no demonstrated synaptic connection from the NTS to ARC POMC and AGRP neurons (Wang et al., Front. Neuroanat., 2015).** Neurons in the spinal cord receive input from sensory neurons in the DRG and send this information to the higher brain areas through the diverse ascending pathways such as the spinothalamic (to the NTS), spinomesencephalic (to the PB), spinoreticular (to the reticular formation), and spinothalamic (to the thalamus) pathways. Hence, although it is highly possible that some AAV-WGA-Cre viruses are taken up by sensory neurons in the periphery, it is very unlikely that WGA-Cre can reach ARC POMC neurons through vagal and/or spinal sensory neuronal transports.

We should emphasize that preganglionic parasympathetic cholinergic neurons directly take up these viruses as there are NO cholinergic postganglionic neurons in the liver. In this case, AAV-WGA-Cre viruses need to cross only one synapse to reach the hypothalamus (i.e. DMV cholinergic neurons to ARC POMC neurons). In our initial experiments to validate our approach, we used AAV8-EF1 α -mCherry-IRES-WGA-Cre instead of AAV8-EF1 α -IRES-WGA-Cre (Ref. Fig. 4). Indeed, these AAV viruses reached the ARC and most liver-projecting neurons in the ARC were POMC neurons (Ref. Fig. 4), consistent with the work from the Friedman group (Stanley et al., PNAS, 2010). In addition to these findings, Figure 2a-e show that the Cre-recombinase delivered by these AAV viruses can successfully drive expression of GFP and Chr2-YFP.

Given all of the above considerations, our approach in addition to being innovative, is the most feasible and, most appropriate for our current study.

Reference Figure 4) Images of confocal fluorescence microscopy showing retrogradely identified POMC neurons.

RE: 1-5. Using the trans neuronal Cre trafficking from the liver in combination with a virus injected in the ARC expressing Chr2 with a Neuronal POMC enhancers (nPEs), the researchers avoided expression of Chr2 in other cells connected to POMC neurons in the ARC. They confirmed the presence of POMC fibers in the cholinergic neurons in the DMV, however, similar to the previous tracing experiment, it is unclear if these projections are exclusive of the DMV or (and most likely) POMC fibers are also present in the surrounding area (i.e. NTS, AP) this is important because the implant was situated above the brainstem, thus all de dorsal vagal complex received light stimulation.

- Yes, it is true that light can activate Chr2-expressing fibers around the DMV. To address this issue, we conducted additional experiments (Suppl. Fig. 1). As shown in Suppl. Fig. 1b, most POMC fibers are found in the DMV, although the NTS also appears to receive POMC input. Interestingly, MC4Rs are densely expressed in the DMV, whereas only a small number of MC4R-expressing cells are scattered in the NTS (Suppl. Fig. 1c). These findings are consistent with the prior findings of the Elmquist group (Liu et al., J. Neurosci., 2003; Kishi et al., J. Comp. Neurol., 2003).

Importantly, activation of MC4Rs inhibits cholinergic neurons in the DMV, which is consistent with other work from the Elmquist group (Sohn et al., Cell, 2013). Moreover, both stimulation of POMC fibers in the DMV and inhibition of liver-projecting cholinergic fibers cause an elevation of blood glucose levels. Knock-down of MC4Rs in the DMV blocks the effect of optogenetic stimulation of the ARC^{POMC}->DMV^{ACh}->liver pathway. Furthermore, optogenetic stimulation of the ARC^{POMC}->DMV^{ACh}->liver projection is no longer able to elevate blood glucose levels *in hepatic-vagotomized mice* (Supplementary Fig. 6b and c). Although we cannot rule out the possibility that Chr2-expressing POMC fibers in the NTS or AP can be activated, we believe that the vagotomy results support our contention that our effects are largely attributable to the ARC^{POMC}->DMV^{ACh}->liver pathway.

RE: 1-6. The authors injected a Cre dependent retroAAV expressing the red-shifted, light-driven inward chloride pump Jaws to silence neuronal activity in Chat cre mice to inhibit hepatic parasympathetic afferents with red light delivered transdermal. Jaws has been reported to cause stronger photocurrents than other opsins. It has been previously reported that an additional consequence of strong ion-pumping activity is an increased firing rate when illumination is abruptly terminated, especially when expressed at high levels. In contrast with the present findings, previous invasive and noninvasive approaches have reported lower levels of glucose after parasympathetic stimulation. Although the authors used a low frequency (1Hz), given the lack of electrophysiological recordings of DMV neurons, it is still possible that the effect observed on glucose could be a side effect of increased firing rate after inhibition.

- A rebound burst of action potentials is common after illumination of cells expressing halorhodopsins or archaerhodopsins, which respectively pump chloride inward and protons outward. A variety of possible mechanisms have been proposed, including hyperpolarization-activated I_h currents or changes in chloride reversal potential due to intracellular chloride accumulation. This predominantly occurs when the cell body itself is illuminated. In contrast, in our case, we used Jaws to block action potential propagation along axons and directly inhibit axon terminals where action potentials arrive. As axon terminals cannot generate action potentials, we do not expect a rebound burst of action potentials in our preparations. As described by the Reviewer, parasympathetic stimulation has been shown to lower blood glucose levels. Thus, increased blood glucose levels following illumination of Jaws-expressing fibers is due to inhibition of parasympathetic nerves, supporting our interpretation that activation of Jaws in axon terminals effectively blocks action potential propagation.

We also performed additional experiments to determine if stimulation of Jaws-expressing cholinergic fibers indeed diminishes ACh levels in liver. **We found a robust reduction in ACh content in the livers of mice injected with Jaws (Fig. 5c)**, further supporting that light illumination inhibits cholinergic fibers.

Re: 1-7. To evaluate a physiological condition that might require activation of the POMC neurons projecting to the liver to increase glucose and counteract hypoglycemia, the researchers followed the activity of the POMC neurons after insulin administration using *in vivo* fiber photometry. In contrast to their previous experiments using the retrograde Cre system, the authors used POMC cre mice and injected a viral vector to express the calcium sensor GCaMP6s in a cre dependent manner, therefore they registered the activity of any POMC neuron, regardless of their connectivity. Typically, hypoglycemia is observed as early as 15 min after insulin administration in non-obese mice, however, POMC activity registered by fiber photometry increased only after 60 min. This delayed response is somehow inconsistent with the fast response observed by optogenetic stimulation (15 min), the authors did not discuss possible explanations for this difference in the response of POMC neurons to hypoglycemia vs the fast effects observed by optogenetic stimulation.

- We completely agree with Reviewer. As described in Re: 3, the initial response to *i.p.* injection of insulin is a decrease in calcium signal. This is highly expected as insulin inhibits the activity in half of ARC POMC neurons. In other words, calcium signal in ARC POMC neurons following *i.p.* injection of insulin appears to be dominated by insulin-induced inhibition of ARC POMC neurons. As described by the Reviewer, insulin-induced hypoglycemia is observed at ~15min and the lowest glucose levels are observed ~60min after insulin treatment (Ref. Fig. 3D). Our newly analyzed results (Fig. 6b) clearly show that *i.p.* injection of insulin induces the biphasic response that consists of the initial insulin-mediated inhibition and the late hypoglycemia-mediated excitation of ARC POMC neurons. Thus, we feel that our observed calcium signal reflects the activity of ARC POMC neurons in response to insulin-induced hypoglycemia.

It is true that we measured calcium signal from the entire population of ARC POMC neurons. It would be great to know if insulin-induced hypoglycemia excites exclusively liver-projecting ARC POMC neurons. In this study, we sought to determine the physiological role of liver-projecting ARC POMC neurons, specifically the $ARC^{POMC} \rightarrow DMV^{ACh} \rightarrow liver$ pathway. To address this, we need to specifically stimulate liver-projecting POMC fibers in the DMV. For this reason, we used AAV8-WGA-Cre and AAV5-nPEs-DIO-ChR2-YFP to express ChR2 exclusively in liver-projecting POMC neurons. Using *in vivo* fiber photometry, we asked if insulin-induced hypoglycemia alters ARC POMC neuron activity. In fact, one cannot expect that hypoglycemia regulates only liver-projecting ARC POMC neurons among other ARC neurons. Hypoglycemia will affect not only ARC POMC neurons but also the neighboring neurons, including AgRP, dopamine, ACh neurons, etc that are located in the mediobasal hypothalamus. Importantly, these neighboring neurons can also regulate ARC POMC neurons activity. Thus, the observed calcium signal is a sum of the effects of insulin, hypoglycemia, and synaptic inputs. Under these conditions, insulin-induced hypoglycemia increases ARC POMC neuron calcium signal. As *i.p.* injection of glucose does not excite ARC POMC neurons (Ref. Fig. 2B), we think that the increased POMC neuron activity is due at least in part to increased ARC POMC neurons innervating to the liver. If so, we expect inhibition of parasympathetic cholinergic neurons, thereby promoting glucose output.

Minor

Re: 1-8. In introduction, the authors mention glucose-lowering effects of POMC neurons (1st paragraph) and then move on to glucose-raising effects of the melanocortin system (2nd paragraph). Here, they link both effects to the sympathetic nervous system, which is confusing.

- In this study, we sought to determine why there are such differences in the literature. Blood glucose levels can be regulated largely by glucose disposal and glucose production. Both are regulated by diverse mechanisms. In this study, we focused on the role of liver-projecting ARC POMC neurons. In fact, POMC, but not AgRP, neurons project to liver (Stanley et al., PNAS, 2010). Activation of MC4Rs inhibits DMV cholinergic

neurons (Sohn et al., Cell, 2013), which is in contrast to the depolarizing effect of MC4Rs in other brain areas. Furthermore, although ARC POMC neurons innervate parasympathetic cholinergic motor neurons, most studies have focused on the roles of melanocortins on the sympathetic nervous system. In our present study, we clearly show that melanocortins have the ability to control the parasympathetic nervous system as well. This novel neural circuit is relatively a simple circuit compared to the sympathetic nervous system regulated by melanocortins (e.g. ARC^{POMC}->PVH->LPB->BST->->->).

Re: 1-9. The authors focused on hepatic gluconeogenesis. However, hepatic glucose production is also from glycogenolysis, as the authors mentioned in introduction. Is there any reason why the authors did not examine the latter?

- We agree with the Reviewer. As described by the Reviewer 2, blood glucose levels can be regulated by reduced hepatic glucose uptake, increased hepatic gluconeogenesis and glycogenolysis. It appears that all these mechanisms may contribute to increased blood glucose levels in our preparations (see Ref. Fig. 6).

Re: 1-10. Results, page 5, first paragraph, ChAT is for choline acetyltransferase, not choline transferase.

- We changed it.

Re: 1-11. Figure 3h-j, labels for x axes are missing.

- We are sorry for that. We labeled x axes.

Re: 1-12. Figure 4e, scrambled (control) shRNA data needs to be added.

- We changed Fig. 4e by adding the results obtained at 1-week post *Mc4r* shRNA injection. In addition to the pharmacological experiments with SHU9119, this additional information supports the interpretation that MC4Rs in the DMV play an important role in regulating hepatic glucose output.

To be honest, it is extremely difficult to perform 4 different surgeries. In our preparations, we injected AAV-WGA-Cre viruses into the liver and waited at least 4-5 weeks. And then AAV-nPEs-ChR2-YFP viruses were injected into the ARC and we waited another 2-3 weeks before implantation of a fiber-optic cannula. For the shRNA experiments, we should inject shRNA viruses into the DMV and implant the fiber-optic cannula into the DMV on the same day. This process took more than 1 hr since 50nl of viruses were injected every 5min. Hence, the survival rate was relatively low. For these reasons, scrambled shRNA was used only for quantification of *Mc4r* gene expression as shown in Supplementary Fig. 4.

Reviewer #2:

Re: 2-1. Please provide the duration of fast for the animals in each of the figure legends. This is important since a 15 hr fast probably emptied the liver of glycogen and this supports the authors' conclusions regarding gluconeogenesis. It is not appropriate to conclude that the plasma glucose level had plateaued at sixty min in Fig. 3. It was clearly still rising. Further, to assume the rise in plasma glucose reflects glucose production is speculative. The liver takes up a small amount of glucose even after a fast when it is in a net production mode. In addition, in the presence of increased glucose and insulin, hepatic glucose uptake is even more likely to occur. The question then arises as to whether the increase in plasma glucose is secondary to an increase in liver glucose production, a decrease in liver glucose uptake or a combination of the two. It would be interesting to know whether glucokinase transcription changed. It would be even more interesting to know if its location in the hepatocyte changed. It seems to me that the best way forward is to conclude that the rise in glucose was a result of an effect on the liver and that could represent either an increase in glucose production or a decrease in glucose uptake or both.

- We changed the sentence. The effect rapidly returned to baseline after optogenetic stimulation. In other words, blood glucose levels rise as long as the neural circuit is stimulated (Ref. Fig. 5).

We are sorry for the confusion. In our experiments, we did not fast the animals. Fasted animals were used only for the PTT and GTT experiments as described in the previous version of the manuscript. We now include this information in the figure legends as well.

In our initial experiments, we actually used fasted animals as baseline blood glucose levels varied from one to another without fasting. Under these experimental conditions, however, only a small number of the mice responded to optogenetic stimulation. This can be due to the lack of glycogen in the liver as noted by the Reviewer and/or of circulating nutrients in fasted animals.

It is possible that glycogenolysis is one of the metabolic pathways that are activated in our experimental conditions. The synthesis and breakdown of glycogen appear to take place simultaneously. Our additional experiments revealed that inhibition of the DMV^{ACh} ->liver pathway increased both liver-specific glycogen synthase 2 (*Gys2*) and glycogen phosphorylase (*Pygl*) mRNA expression in **male** mice (Ref. Fig. 6A), suggesting that glycogenolysis and glycogenesis may occur. To further address this issue, we measured glycogen content in the liver with and without light illumination and found that there was no significant difference in glycogen content between the two groups (Ref. Fig. 6B and D). Interestingly, female mice exhibited **NO** changes in *Gys2* and *Pygl* mRNA expression and in liver glycogen content (Ref. Fig. 6C). These results suggest that there exist several cellular mechanisms that are activated by inhibition of parasympathetic cholinergic input to the liver.

Reduced glucose uptake can also increase blood glucose levels, as suggested by the Reviewer. To examine this possibility, we conducted additional experiments by analyzing liver-specific glucose transporter 2 (*Slc2a*) and glucokinase (*Gck*) mRNA expression. Although we found no significant difference in *Slc2a2* mRNA expression in the liver (Ref. Fig. 6A and C), it appears that higher increase in blood glucose is associated with lower *Slc2a2* mRNA expression in **female** mice (Ref. Fig. 6E). As female mice showed no difference in *Gys2* and *Pygl* mRNA expression, decreased glucose uptake would be a possible mechanism that elevates blood glucose levels in female mice. *Gck* mRNA expression was upregulated in both male and female mice.

We would like to remind you that optogenetic stimulation of the ARC^{POMC} -> DMV^{ACh} pathway and of the DMV^{ACh} ->liver pathway increased *G6pase* and *Pepck* mRNA expression in **both male and female mice** unlike other enzymes (Ref. Fig. 6A and C). Moreover, our additional experiments further revealed that alanine aminotransferase 1 and 2 (*Gpt1* and 2) mRNA expression was robustly upregulated in male mice (Ref. Fig. 6A and B). It has been reported that alanine aminotransferase is a downstream target of muscarinic receptors in hepatocytes (Vatamaniuk et al., Life Sci., 2003). Given that the expression of gluconeogenic enzymes are subject to modulation by muscarinic receptors in male and female mice, we think that elevated blood glucose levels would be explained at least in part by increased hepatic gluconeogenesis in **both male and female mice**. This possibility was further supported by the PTT experiments in fasted animals. However, as indicated by the Reviewer, increased blood glucose levels following inhibition of the parasympathetic nervous system

would be due to diverse cellular mechanisms, including reduced glucose uptake, increased glycogenolysis, and gluconeogenesis. Accordingly, we revised our discussion (page 17-18).

Reference Figure 6) Changes in mRNA expression for glucose metabolism in the liver with and without inhibition of parasympathetic cholinergic input to the liver. **A** and **B**. *G6pase* and *Pepck* mRNA expression was upregulated in both male and female mice. A robust increase in alanine aminotransferase 1 and 2 (*Gpt1* and *2*) mRNA expression was observed only in male mice. In addition, males exhibited an increase in *Gys2* and *Pygl* mRNA expression. (male, w/o Jaws, n=7-8 mice, w Jaws, n=13-15 mice; female, w/o Jaws, n=7-10 mice, w Jaws, n= 11-13 mice; Unpaired t-test, *p<0.05, ***p<0.001)

C and **D**. No change in glycogen content in male and female mice. **E**. Plot showing relative *Slc2a2* mRNA expression vs. changes in blood glucose levels.

F. Schematic illustration showing changes in mRNA expression for glucose metabolism in male and female mice. light pink circle: genes upregulated in both male and female mice; light blue circle: genes upregulated only in male mice.

We did not include these additional data in the revised version of the manuscript. In fact, although the effect of inhibition of parasympathetic cholinergic input to the liver is the same in both males and females, it seems likely that the cellular mechanisms that mediate this increase are different in male and female mice. Hence, we feel that it is too premature to include these data in the manuscript as there is still much work to be done.

2. The optogenetic increases in mRNA for G6Pas and PEPCK at 1 hr were not large and it seems unlikely that the protein levels of these enzymes would have been altered at sixty min, let alone after the first 15 min when the initial rise in glucose occurred. Did the authors ever measure glycogen in the liver to ensure that they were depleted? Such would be expected after a 15 hr fast but it would be reassuring to know that it were actually measured.

- As suggested by the Reviewer, we performed additional experiments to measure glycogen content in liver (Ref. Fig. 6C and D). There was no significant difference between the control and stimulation groups in male and female mice.

Molecular mechanisms that control hepatic glucose metabolism are divided into two categories: acute and long-term. The acute effects are brought by changes in metabolic flux controlled by protein modifications or allosteric effectors and the long-term effects are mediated by changes in the mRNA expression of key enzymes for glucose metabolism. The activity of gluconeogenic enzymes is catalyzed by specific enzymes and is subject to modulation by hormones and neurotransmitters such as acetylcholine (Hampson and Agius, FEBS letters, 2007). In addition, the activity of alanine aminotransferase that convert alanine into pyruvate is regulated by muscarinic receptors in hepatocytes (Vatmaniuk et al., Life Sci., 2003).

In addition, the activity of glycogen synthase and glycogen phosphorylase are also regulated by phosphorylation. In fact, acute treatment with ACh results in inactivation of glycogen phosphorylase, stimulation of glycogen synthesis in *in vitro* hepatocyte preparations (Hampson and Agius, FEBS letters, 2007). Hence, inhibition of parasympathetic cholinergic fibers would cause the opposing effects such as activation of glycogen phosphorylase and inactivation of glycogen synthesis.

AMPK activates glycogen phosphorylase and inactivate glycogen synthase, as well as inhibit glucokinase translocation and glucose phosphorylation in hepatocytes (for review, Moore et al., Adv. Nutr., 2012). It appears that M3 receptors have the ability to regulate AMPK in hepatocytes (Jadeja et al., Biochem. Pharmacol., 2019) and glucose uptake in L6 skeletal muscle cells through AMPK (Merlin et al., Cell. Signal. 2010).

All these prior studies suggest that the initial rise in blood glucose levels would be explained by increased activity of key enzymes in gluconeogenesis and glycogenolysis in addition to reduced glucose uptake. Increased mRNA expression of these enzymes by optogenetic stimulation would play a role in generating these key enzymes, thereby maintaining elevated blood glucose levels.

RE: 3 and 4. Given the rate of rise in glucose, glucose production must have exceeded glucose utilization for the full hour of stimulation even though glucose utilization must have risen due to increases in both glucose and insulin. If gluconeogenesis explains the rise in glucose production, where did the substrates come from? The increase in glucose and insulin would have decreased lipolysis and reduced the flow of FFA and glycerol to the liver. Production of amino acids by muscle would not have been expected to increase leaving lactate as the only carbon source. Did the authors' measure lactate? Further, what would the site of allosteric regulation be such that it could activate quickly and produce a rise in plasma glucose in 15 min? (4) Is there any chance that plasma epinephrine was increased. This would have caused a large flux of lactate from muscle to the liver, which in turn may have increased glucose production. It would also have inhibited glucose uptake by muscle possibly explaining at least in part the ensuing hyperglycemia.

- As suggested by the Reviewer, we performed additional experiments. We found that inhibition of parasympathetic cholinergic input did not change plasma lactate levels in both male and female mice (Supplementary Fig. 8). We also measured plasma epinephrine levels and found that there was no significant change in epinephrine levels in male and female mice (Supplementary Fig. 8). Thus, it appears that lactate from muscle to liver is not a substrate for gluconeogenesis in our preparations.

It has been described that muscarinic receptors in hepatocytes regulate the activity of alanine aminotransferase (Vatamaniuk et al., Life Sci. 2003), which opens the possibility that circulating amino acids would be a substrate for gluconeogenesis. Interestingly, *Gpt1* and *Gpt2* mRNA expression was robustly upregulated in male mice, which may support this possibility.

As noted by the Reviewer, the value of the paper lies in the link between the liver and the ARC. The exact mechanism, in which liver is increasing glucose levels, remains to be clarified. We will seek to determine the exact cellular mechanisms underlying increased blood glucose levels in our future work.

5. The intraperitoneal GTT data are interesting but the difference in plasma glucose is primarily due to a decrease in hepatic glucose uptake rather than increase in hepatic glucose production. The combination of IP glucose delivery,

hyperglycemia and hyperinsulinemia can cause the liver to take up glucose at very high rates. This would suggest that normal parasympathetic input to the liver is important to glucose tolerance as suggested by Shimazu et al. many years ago. Did you measure insulin levels in the glucose tolerance experiments? Presumably, it would be higher in the laser on group making the finding even more compelling. You should also keep in mind that the level of insulin at the liver is 3 times higher than it is in peripheral blood.

- We agree, and thank the Reviewer for these constructive suggestions. As described above, a rise in blood glucose can be explained in part by reduced glucose uptake.

There is no doubt that parasympathetic cholinergic input to the liver can play an important role in glucose tolerance by increasing glucose uptake. In this study, we focused on the effects of inhibition of parasympathetic cholinergic neurons on blood glucose levels. In fact, activation of MC4Rs inhibits vagal cholinergic neurons, while exciting sympathetic cholinergic neurons (Sohn et al., Cell, 2013). Most studies have examined the role of melanocortins in the control of the sympathetic nervous system, although parasympathetic cholinergic neurons are also a downstream target of the central melanocortin system. Our current study clearly demonstrates that inhibiting parasympathetic input to the liver is able to increase blood glucose levels. This mechanism would be important for counterregulatory response to insulin-induced hypoglycemia.

During the GTT experiments, we were not able to collect blood samples. Collecting blood samples from the retroorbital plexus in awake mice every 15 min was not feasible. In our initial experiments, we tried to collect blood samples every 15min from anesthetized mice to analyze plasma hormone levels, while stimulating the ARC^{POMC}->DMV^{ACh} pathway. In fact, this information can give us some clues about how melanocortins regulate blood glucose levels. However, this induced too much stress even in anesthetized mice.

6. There are numerous places in the discussion where the plasma glucose change has been attributed to glucose production. The authors should either measure glucose production or they should take a more cautious approach to interpreting the data. The value of the paper lies in the link between the liver and the ARC. The exact mechanism, in which liver is increasing glucose levels, remains to be clarified.

- While revising the manuscript, we realize that, as noted by the Reviewer, a rise in blood glucose levels can be attributed to reduced glucose uptake, increased gluconeogenesis and/or glycogenolysis. With our present data, we cannot conclude that elevated blood glucose is solely due to glucose production. Thus, we revised the text throughout the manuscript. We appreciate your thoughtful comments.

Reviewer #3:

1. The authors take an approach of combined retrograde Cre transport from liver with a POMC enhancer and Cre dependent expression of ChR upon AAV injection in the ARC. Although this is a potentially elegant approach to specifically target liver innervating POMC neurons, it should be complemented to stimulating DMV-projecting fibers in POMC Cre mice injected with a Cre-dependent ChR-expressing virus in the ARC.

- In our initial experiments, we actually used the POMC-Cre mice injected with a Cre-dependent ChR-expressing virus into the ARC. The results obtained from these initial experiments are shown in Ref. Fig. 7. Optogenetic stimulation of ChR-expressing POMC fibers in the DMV induces no effect or increases or decreases blood glucose levels. In fact, the DMV controls diverse peripheral organs, including the gut, pancreas, liver, and heart, etc. The neurons in the DMV express MC4Rs as well as μ -opioid receptors (MORs) (Browning et al., J. Neurosci., 2002; Duan et al., Brain Res. Bull., 1990) that are activated by the neuropeptides released from ARC POMC neurons. Accumulated studies have demonstrated that ARC POMC neurons are neurochemically, anatomically and functionally heterogeneous (Jeong et al., PLoS One, 2016; Lee et al., Nat. Comm., 2015; Elias et al., Neuron, 1998; King and Hentges, PLoS One, 2011; Koch et al., Nature, 2015; Wei et al. PNAS, 2018). The roles of POMC neuron heterogeneity have not been well explored due to the lack of appropriate experimental approaches permitting selective stimulation of neuroanatomically-identified POMC neuronal subpopulations. For example, acute stimulation of the entire ARC POMC population does not produce the canonical reduction in food intake (Zhan et al., J. Neurosci., 2013; Aponte et al., Nat. Neurosci., 2011; Fenselau et al., Nat. Neurosci., 2017; Uner et al., Sci. Rep., 2019), despite the well-known anorexigenic effects of α -MSH. Interestingly, activation of ARC POMC neurons has been shown to promote food intake through the release of opioids (Koch et al., Nature, 2015; Wei et al., PNAS, 2018). Our prior study further shows that optogenetic stimulation of TRPV1-expressing POMC neurons reduces food intake via MC4R activation (Jeong et al., PLoS Biol., 2018). Given ARC POMC neurons are heterogeneous and DMV neurons express both MC4Rs and MORs, it is expected that stimulation of ChR-expressing POMC fibers in the DMV would produce such different effects on blood glucose levels.

To specifically address this important issue, we developed our approach. As there are no postganglionic parasympathetic neurons in the liver, parasympathetic cholinergic motor neurons can directly take up AAV-WGA-Cre viruses. Importantly, WGA is preferentially transported in a retrograde direction in the peripheral nervous system (Sugita and Shiba, Science, 2005; Ohmoto et al., Mol. Cell. Neurosci., 2008; Bai et al., Cell, 2019, etc). Moreover, WGA is among the first genetically targeted transsynaptic tracers from the PNS to the CNS (Horowitz et al., PNAS, 1999). Under these experimental conditions, AAV-WGA-Cre viruses need to cross only one synapse to reach the hypothalamus (i.e. DMV cholinergic neurons to ARC POMC neurons). Although it appears that most liver-projecting neurons in the ARC are POMC neurons (Stanley et al., PNAS, 2010), we further developed AAV having the neuronal POMC enhancers (identified by the Low and Rubinstein group) to express ChR exclusively in liver-projecting POMC neurons. Under these experimental conditions, no mice showed a decrease in blood glucose levels during optogenetic stimulation of the ARC^{POMC}->DMV^{ACH} pathway. Hence, we feel that our approach in addition to being innovative, is the most feasible and, most appropriate for our current study.

2. This is particularly important, since the present study is in striking contrast to earlier work by the Elmquist lab, showing that selective re-expression of the MC4R in Chat-expressing neurons in the DMV IMPROVES glucose metabolism and SUPPRESSES hepatic glucose production (Rossi et al., Cell Metab. 2011). The authors of the present manuscript incorrectly refer to this study, when they state on page 15 "Moreover, there is no improvement in hyperglycemia that is observed in obese loxTB MC4R mice". As shown in Fig. 5 of the study by Rossi et al., selective re-expression of MC4Rs in Chat neurons, reduces blood glucose concentrations (Fig. 5a), plasma insulin concentrations (Fig. 5b), improves GIR during a clamp (Fig. 5e) and most importantly almost completely normalizes insulin-induced suppression in HGP (Fig. 5g). There is no solution offered for this striking discrepancy.

- We are sorry for the confusion. Indeed, the Elmquist research group demonstrates that selective re-expression of MC4Rs in cholinergic neurons in ChAT-Cre; loxTB MC4R mice significantly reduces blood glucose concentrations and plasma insulin concentrations comparing with Lox-Tb MC4 mice. However, *re-expression of MC4Rs exclusively in preganglionic parasympathetic neurons in Phox2b mice shows no changes in blood glucose (Fig 5C), HGP (Fig 5H), GIR, and glucose disposal (Fig 5J)*. Based on these findings, they clearly state in their discussion that *“the selective re-expression of MC4Rs in both cholinergic preganglionic sympathetic and parasympathetic neurons reduces hyperglycemia and hyperinsulinemia, without changes in the feeding behavior. While hyperglycemia was clearly improved in ChAT-Cre, loxTB MC4R mice, this was NOT observed in Phox2b-Cre, loxTB mice. Thus, our data suggest that the effect of melanocortins to suppress blood glucose levels may be at least in part mediated through MC4Rs expressed in preganglionic sympathetic neurons. However, we cannot exclude the possibility that the improvements in hyperglycemia seen in ChAT-Cre, loxTB MC4R mice might be partially indirect, and mediated by improved body composition”*. (Rossi et al., Cell Metab. 2011). Hence, the work by the Elmquist group does NOT conclude that re-expression of MC4Rs in parasympathetic cholinergic neurons changes blood glucose levels. We should also emphasize that they use the loxTB strain that lacks MC4Rs in the whole body. As not only MC4Rs in cholinergic neurons but also in other brain areas can regulate energy intake and expenditure (Balthasar et al, Cell, 2005), the lack of MC4R expression in other areas can change glycemia.

Importantly, several studies have described the importance of central cholinergic neurons in the regulation of energy metabolism. For instance, cholinergic neurons in the basal forebrain suppress food intake via ARC POMC neurons and ablation of these cholinergic neurons causes increased body weight, leptin, and insulin levels and reduces ARC POMC expression (Herman et al., Nature, 2016). Cholinergic neurons in the DMH are strongly involved in the regulation of BAT thermogenesis and food intake (Jeong et al., 2015 Molecular metabolism, Jeong et al., 2017, Molecular metabolism). It is also reported that the ARC contains cholinergic neurons (Meister et al., 2006, European J. Neurosci.; Jeong et al. 2016, Plos One). Among the neurons in the ARC, a subset of AgRP and POMC neurons express cholinergic neurons and these cholinergic neurons appear to regulate food intake (Calarco et al., 2018, European J. Neurosci). In other words, the experimental approaches used by the Elmquist group (ChAT-IRES-Cre X LoxTB MC4R) do not guarantee re-expression of MC4Rs in cholinergic neurons exclusively in the autonomic nervous system (ANS). Rather, they re-express MC4Rs in cholinergic neurons throughout the brain. Furthermore, expression of MC4Rs described above would be able to change energy metabolism and glucose homeostasis. In our present study, however, we focused on liver-projecting cholinergic neurons in the DMV among other cholinergic neurons in the ANS and brain. Hence, we do NOT think that there is a striking discrepancy between the work by the Elmquist group and by our group.

Rather, we feel that there is a consistency between the two studies. DMV cholinergic neurons express MC4Rs and activation of MC4Rs in the DMV inhibits DMV cholinergic neurons (Sohn et al., Cell, 2013). It is known that activation of the sympathetic nervous system increases hepatic glucose production, whereas activation of the parasympathetic nervous system induces the opposing effect. The study of the Elmquist group (Cell, 2013) clearly demonstrates that MC4Rs oppositely regulate sympathetic and parasympathetic preganglionic neurons. Thus, we specifically sought to define the role of this inhibitory effect of MC4R as, to our best knowledge, this inhibitory effect of MC4Rs is unique as activation of MC4Rs excites neurons in most cases. In our present study, we provide neurophysiological evidence that inhibition of parasympathetic cholinergic input to the liver elevates blood glucose levels.

3. The presented Ca imaging studies are not impressive. First, the technical description how they were performed and analyzed is minimal. Second, the quantitative effects are marginal, i.e. 1% difference in dF/F comparing saline and insulin at 90 minutes after insulin injection. Third, there is no rationale provided for this kinetic. Upon injection of the used dose of insulin hypoglycemia should be apparent after 30 minutes (no data are provided for this, but based on earlier studies (Tooke et al., Mol. Metab. 2019)). Strikingly, Tooke et al., report a robust induction of Fos mRNA expression in POMC neurons 30 minutes after an injection of the same dose of insulin. There is no reason, why direct assessment of Ca activity should occur only 60 minutes after this event. In fact it is predicted to occur prior to changes in Fos expression, if anything meaningful was measured.

- We now provide a full description of how we performed and analyzed *in vivo* fiber photometry (page 25-26). We agree, and thank the Reviewer for these constructive suggestions. We regraphed Fig. 6 to clarify the differences we observed. This new figure shows a clear difference between the two groups.

Indeed, the difference in calcium signal between the control and treatment groups is small. However, we should emphasize that the difference in calcium signal *in ARC POMC neurons* is generally quite small. For example, the work from the Betley lab (Alhadeff et al. *Neuron*, 2019) shows changes in calcium signal in ARC POMC in response to different substances (Ref. Fig. 2A and B). Changes in calcium signal in ARC POMC neurons range from 0.2% to 0.5% $\Delta F/F$. This would be due in part to the fact that this assessment reflects **the summed activity** of the entire population of ARC POMC neurons. Multiple, accumulating studies have described that ARC POMC neurons are heterogeneous in the context of the neurotransmitters/ neuropeptides, the receptors for hormones and nutrients, and the neurophysiological responses. The altered POMC neural activity is a sum of this heterogeneous POMC neuron activity. For instance, 10% of ARC POMC neurons are excited, while half of them are inhibited by insulin. A third of POMC neurons do not respond to insulin (Dodd et al., *Elife*, 2018; Qiu et al., *Cell Metab.*, 2014). The heterogeneous neurophysiological responses of ARC POMC neurons to neurotransmitters (Sohn et al., *Neuron*, 2011), hormones (Lee et al., *Nat. Comm.*, 2015), and nutrients (Patron et al., *Nature*, 2007) are very common. Thus, the **net** neural activity expressed as $\Delta F/F$ may in fact be expected to be small.

Yes, it is true that *c-fos* mRNA expression can be detected 30 min post i.p. injection of insulin (Tooke et al., *Mol. Metab.*, 2019). However, it is not clear if this increased activity is due to insulin-induced hypoglycemia or insulin itself. In fact, as described above, insulin either excites or inhibits ARC POMC neurons (Dodd et al., *Elife*, 2018; Qiu et al., *Cell Metab.*, 2014). In our preparations, i.p. injection of insulin initially reduced POMC neuron activity, consistent with insulin-mediated inhibition of POMC neurons. In fact, half of ARC POMC neurons are inhibited (Dodd et al., *Elife*, 2018). This insulin's inhibitory effect appears to be reversed by insulin-mediated hypoglycemia as the lowest glucose levels were observed 1 hr post i.p. insulin injection in our preparations. The biphasic response would be attributable to the initial insulin-mediated inhibition followed by the hypoglycemia-mediated excitation of ARC POMC neurons. Interestingly, i.p. injection of glucose does not depolarize ARC POMC neurons (Ref. Fig. 2B, blue arrow), which is in contrast with the neurophysiological response of ARC POMC neurons to glucose (Patron et al., *Nature*, 2007; blue arrow). The results obtained from individual POMC neurons and the entire POMC neurons appear to be different. Thus, we feel that our observed calcium signal reflects the activity of ARC POMC neurons in response to insulin-induced hypoglycemia.

Reference Figure 3) A and B. Changes in calcium signal in ARC POMC neurons in response to different substances. The responses ranged from 0.2% to 0.5%. Note that i.p. injection of glucose induced **NO** effect (blue arrow). Figures were taken from the work by the Betley group (Alhadeff et al. *Neuron*, 2019). **C and D.** i.p. injection of insulin induced the biphasic response. Pooled data from 5 mice showing changes in blood glucose levels post i.p. injection of insulin (D). This figure is the same as shown in page 2.

4. The YFP-positive POMC fibers in Fig. 1g, h are barely visible. Moreover, co-staining for endogenous α -MSH should be performed to characterize them as bona fide POMC neuron projections.

- We performed additional experiments and changed Fig. 2g and h to better show the YFP-positive fibers and terminals in the DMV. We also included a new figure as supplementary figure (Suppl. Fig. 2), showing that most YFP-positive neurons are indeed POMC neurons. Light illumination also caused pS6 protein (a neuronal activity marker) in ARC POMC neurons in C57BL/6J mice injected with AAV-WGA-Cre and AAV-nPE-DIO-ChR2-YFP viruses (see Ref. Fig. 1). We also would like to emphasize that our constructs have been successfully used to study TRPV1-expressing POMC neurons (Jeong et al., PLOS Biology, 2018). This prior study further validates the expression and efficiency of our constructs.

Although we have tried to label POMC neurons and fibers with anti- α -MSH (Millipore, AB5087) and anti-ACTH (Abcam, ab74976; Santa Cruz, sc-57021) antibodies multiple times, unfortunately we could not get good signal in POMC cell bodies and fibers. This appears to be common (Wittmann et al., J. Comp., Neurol., 2017). Importantly, α -MSH staining works only after treatment with colchicine (Balthasar et al., Neuron, 2004). Colchicine blocks axonal transport, so peptides and proteins concentrate in cell bodies, not fibers. We should emphasize that we used C57BL/6J mice and injected AAV-WGA-Cre into the liver and AAV-nPE-DIO-ChR2-YFP into the ARC. These enhancers have been shown to drive reporter gene expression exclusively to POMC neurons in the ARC (de Souza, Mol. And Cell. Biol., 2005; Lam et al., PLOS Genetics, 2015). Moreover, our new data (Suppl. Fig. 1) show that DMV cholinergic neurons express MC4Rs and receive direct POMC input from the ARC. Hence, we feel that the observed POMC fibers in the DMV are indeed POMC projections.

Reference Figure 1) Images of confocal fluorescence microscopy showing expression of pS6 protein a neuronal activity marker in ARC POMC neurons following light illumination. **A.** C57BL/6J Mice were injected with AAV5-nPE-DIO-ChR2-YFP only. **B.** C57BL/6J Mice were injected with both AAV5-nPE-DIO-ChR2-YFP and AAV8-WGA-Cre. The ARC was illuminated with light in both cases. White arrows indicate co-expression of pS6 and POMC in ARC POMC neurons.

Reviewers' Comments:

Reviewer #1:

Remarks to the Author:

The authors have responded to most of the previous criticisms. Laudably, they include a significant amount of new data. I have no further concerns.

Reviewer #2:

Remarks to the Author:

General comments

The revision has improved the manuscript significantly. The caution used in interpreting the cause of the rise in glucose is appropriate. As noted by the authors, it points to the need for an additional study addressing the mechanism by which the elevation in glucose is brought about.

Specific comments

1. Line 160. Change the text to state whether the rise in blood glucose was associated with increased gluconeogenic enzyme expression rather than increased hepatic glucose production. This makes the theme consistent throughout the text.

2. Without enzyme levels, I remain doubtful that the increased expression of G6Pa's and PEPCK caused the increase in plasma glucose. The time course is just not reasonable. Further, if the liver glycogen levels after a 15 hr fast were 7/8 mg/gm (see ref fig 6), the animals were not totally free of liver glycogen despite the fast. To further my point, the difference in the groups in response to the PTT at 60 min was small (40 mg/dl) despite a large bolus of gluconeogenic substrate. Since the authors no longer try to explain the rise in glucose as a function of gluconeogenesis, I am comfortable with their position. Clearly it would be interesting to understand more about the explanation for the rise in blood sugar.

3. Please add a note regarding the duration of fast of the animals in Figure 5 and where appropriate in the supplementary figures.

Reviewer #4:

Remarks to the Author:

I was very happy reading the revised manuscript from Kwon et al, and their excellent responses to the last round reviewers' comments. I agree with the comments raised by the prior reviewers and I also feel that the current revised manuscript with additional data has been dramatically improved and the conclusions have also been greatly strengthened.

My minor concern is that, like what Reviewer-1 had commented on the retrograde tracing experiments from the liver, the use of AAV-WGA is still potentially a problem, even after the authors demonstrated now successful labeling of some ARC neurons. Since a mCherry-WGA virus has been used, I'd like to suggest the authors include some additional supplemental data showing some other brain regions with mCherry labeling with verified neurons innervating the liver, and some brain regions without mCherry and with neurons that are known not talking with the liver. These additional data are expected to be great control results.

We thank the Editor and expert Reviewers for their careful and detailed review of the revised manuscript.

REVIEWER COMMENTS

Reviewer #2:

Re: 1. Line 160. Change the text to state whether the rise in blood glucose was associated with increased gluconeogenic enzyme expression rather than increased hepatic glucose production. This makes the theme consistent throughout the text.

- We revised the sentence (page 8).

2. Without enzyme levels, I remain doubtful that the increased expression of G6Pa's and PEPCK caused the increase in plasma glucose. The time course is just not reasonable. Further, if the liver glycogen levels after a 15 hr fast were 7/8 mg/gm (see ref fig 6), the animals were not totally free of liver glycogen despite the fast. To further my point, the difference in the groups in response to the PTT at 60 min was small (40 mg/dl) despite a large bolus of gluconeogenic substrate. Since the authors no longer try to explain the rise in glucose as a function of gluconeogenesis, I am comfortable with their position. Clearly it would be interesting to understand more about the explanation for the rise in blood sugar.

- We thank the Reviewer for the constructive suggestion. Yes, we will seek to understand more about the explanation for the rise in blood glucose in our future study.

3. Please add a note regarding the duration of fast of the animals in Figure 5 and where appropriate in the supplementary figures.

- We now added the duration of fast of the animals in Fig. 5 and Supplementary Fig. 3 and 5.

Reviewer #4:

Re: My minor concern is that, like what Reviewer-1 had commented on the retrograde tracing experiments from the liver, the use of AAV-WGA is still potentially a problem, even after the authors demonstrated now successful labelling of some ARC neurons. Since a mCherry-WGA virus has been used, I'd like to suggest the authors include some additional supplemental data showing some other brain regions with mCherry labelling with verified neurons innervating the liver, and some brain regions without mCherry and with neurons that are known not talking with the liver. These additional data are expected to be great control results.

- We thank the Reviewer for the constructive suggestion. We used the mCherry-WGA-Cre virus only in our initial experiments to know whether our virus can reach the hypothalamus. Since then, we removed the mCherry reporter protein as described in the Methods section and used AAV8-EF1 α -IRES-WGA-Cre viruses to perform double labeling with anti-GFP and anti-POMC antibodies.

Our viral vector contains the WGA-Cre fusion protein. In other words, we are able to deliver the Cre recombinase gene via axonal transport. As described in our manuscript, we determined transneuronal transport of this fusion protein and its Cre recombinase activity. We found that direct injection of AAV8-WGA-Cre into the liver of floxed-stop Rosa26-GFP mice induced Cre-mediated expression of GFP *exclusively* in the ARC (see Figure; white circle), supporting the contention that the Cre recombinase is successfully delivered to the ARC.

We should also emphasize that we administered an AAV5 encoding Cre-dependent channelrhodopsin-2 (ChR2) and enhanced yellow fluorescent protein (eYFP) fusion protein under control of the two neuronal POMC enhancers (nPEs) (AAV5-nPE-DIO-ChR2-YFP) into the ARC of C57BL/6J *wild-type* mice to *exclusively* control the activity of liver-projecting POMC neurons in the ARC. As shown in the figure, this also induced Cre-mediated expression of YFP *exclusively* in the ARC (white circle). Based on these findings, we strongly believe that our approach is appropriate for our study.